# Rules warp feature encoding in decision-making circuits

**R. Becket Ebitz** *, **Jiaxin Cindy Tu**, **Benjamin Y. Hayden**

Department of Neuroscience, Center for Magnetic Resonance Research, and Center for Neuroengineering
University of Minnesota, Minneapolis, Minnesota, United States of America

* rebitz@gmail.com

## Abstract

We have the capacity to follow arbitrary stimulus–response rules, meaning simple policies that guide our behavior. Rule identity is broadly encoded across decision-making circuits, but there are less data on how rules shape the computations that lead to choices. One idea is that rules could simplify these computations. When we follow a rule, there is no need to encode or compute information that is irrelevant to the current rule, which could reduce the metabolic or energetic demands of decision-making. However, it is not clear if the brain can actually take advantage of this computational simplicity. To test this idea, we recorded from neurons in 3 regions linked to decision-making, the orbitofrontal cortex (OFC), ventral striatum (VS), and dorsal striatum (DS), while macaques performed a rule-based decision-making task. Rule-based decisions were identified via modeling rules as the latent causes of decisions. This left us with a set of physically identical choices that maximized reward and information, but could not be explained by simple stimulus–response rules. Contrasting rule-based choices with these residual choices revealed that following rules (1) decreased the energetic cost of decision-making; and (2) expanded rule-relevant coding dimensions and compressed rule-irrelevant ones. Together, these results suggest that we use rules, in part, because they reduce the costs of decision-making through a distributed representational warping in decision-making circuits.

## Introduction

We have a tremendous capacity to change how we perceive and respond to the world as our environment and goals change. One central part of this flexibility is our ability to learn rules: simple policies for guiding behavior that allow us to make decisions quickly, reasonably accurately, and with little subjective sense of effort [1]. We know from many studies that rule identity is encoded in firing rate changes of neurons in specific brain regions, classically in the dorsolateral prefrontal cortex (dlPFC; [1–4]), but also in regions implicated in decision-making, like the orbitofrontal cortex (OFC) and striatum [5–10]. However, implementing a rule requires more than simply encoding its identity in the right structures, and an important question remains unanswered: How do rules shape neural processing?

One possibility is that rules warp the way the world is represented in decision-making circuits. Many decisions require us to solve very high-dimensional problems, where the correct response is some complex function of our past choices, reward history, and the various

**Data Availability Statement:** All data are available on figshare (https://doi.org/10.6084/m9.figshare.13139450.v1).

**Funding:** Support provided by the National Institute on Drug Abuse (BYH: R01-DA038106), the Brain & Behavior Research Foundation (RBE: Young

Investigator Award #27298), and a Momental Foundation Unfettered Research Grant to RBE (https://www.momentalfound.org/). The funders had no role in study design, data collection and analysis, decision to publish, or preparation of the manuscript.

**Competing interests:** The authors have declared that no competing interests exist.

**Abbreviations:** AIC, Akaike information criterion; BIC, Bayesian information criterion; BOLD, blood-oxygen-level-dependent; CSST, Cognitive Set-Shifting Task; dlPFC, dorsolateral prefrontal cortex; DS, dorsal striatum; fMRI, functional magnetic resonance imaging; HMM, hidden Markov model; OFC, orbitofrontal cortex; STD, standard deviation; STE, standard error; VS, ventral striatum; WCST, Wisconsin Card Sorting Task.

features of the options available to us in the moment [11]. However, computation takes time and energy [12], especially when performed with metabolically expensive spikes [13,14]. Thus, in practice, we use various strategies to simplify decision-making [15–17]. For example, we may execute a win–stay/lose–shift strategy where we only need to keep track of the last choice and its outcome, then compare new options against the last choice according to the rule determined by the last outcome. There is no need to calculate the value of each alternative or to represent information acquired further back into the past. Stimulus–response rules could simplify decision-making even more. To decide based on a well-learned stimulus–response rule, one only needs to determine whether or not an option satisfies the rule, which could be as simple as picking a berry that has the right color. This eliminates the need to represent or perform computations on any rule-irrelevant information and could thereby reduce the energetic and metabolic costs of decision-making [11,15,16,18,19].

Although stimulus–response rules could simplify decision-making in theory, it is not clear if the brain can take advantage of this opportunity. One influential hypothesis is that the brain implements rules and other executive functions as a kind of attentional gate that enhances the representation of relevant features of the world at the expense of irrelevant features [20]. Eliminating the early sensory encoding of irrelevant information would certainly reduce the costs of decision-making. However, tests of this hypothesis in both prefrontal cortex [21,22] and early sensory cortex [23,24] have failed to find evidence of early feature selection. Information about irrelevant features of behaviorally relevant stimuli is still strongly encoded in these regions, even when it can interfere with task performance [4,21,22]. Further, while feature-based attention can enhance the responses of sensory neurons that are tuned to prefer attended features [25–31], this does not mean that feature-based attention can selectively enhance a neuron's response to 1 feature of a stimulus but not another. In fact, attending to 1 feature of a stimulus may enhance information about its other features, even when they are behaviorally irrelevant [23,32–34]. If we are trying to pick the ripest berry, can we really choose only based on color in order to take advantage of the computational efficiency that could be gained through ignoring shape and size?

Here, we build on theoretical work that suggests that executive processes like rules may alter how sensory information is transformed into motor responses [22,35]. We focus on measuring population representations of choice features in 3 areas implicated in decision-making and reward processing: Area 13 of the OFC, the nucleus accumbens core of the ventral striatum (VS), and both the caudate and putamen in the dorsal striatum (DS). We have long been interested in the role of structures linked to reward in cognitive processes and motivated by a belief that the distinction between motivational and cognitive processes is overly simplistic. For this reason, we were especially interested in looking at how rule-based decision-making is implemented in core reward regions, like OFC and VS, and in regions implicated in both reward processing and motor control/learning, like DS.

Previous studies have typically examined the neural basis of rules by comparing periods in which 1 rule or another is imposed by the task [1–4,7,22,36–38]. These studies then define rule-based decisions as the correct decisions that occur once rules have been learned. However, here, we took a computational approach, modeling rules as the latent stimulus–response policies underlying the monkeys' decisions [39]. This allowed us to the neural correlates of these latent rule states, as well as to contrast rule-based decisions with decisions that could not be explained by simple stimulus–response rules.

# Results

Two rhesus macaques performed a total of 128 sessions (73,627 trials) of the Cognitive Set-Shifting Task (CSST, **Fig 1A**, [7,40]), a macaque analog of the Wisconsin Card Sorting Task

(WCST) that encourages subjects to discover and apply hidden stimulus–response rules to make decisions. On each trial, a unique combination of the 3 colored (cyan, magenta, and yellow) shapes (circle, star, and triangle) appeared in random order at 3 screen locations. On each trial, there was a correct color or shape (six possible "correct features"). Only choices that matched the correct feature were rewarded. The correct feature was fixed for a block of 15 correct trials, then it changed and a new correct feature was chosen at random (See Methods). When the subjects discovered the correct feature, they had the opportunity to follow a rule where they only chose options that matched the correct feature, generating a sequence of choices that shared a single color or shape feature (**Fig 1B**). However, after the correct feature changed, subjects had to discard any rule they had been following in order to discover the new correct rule.

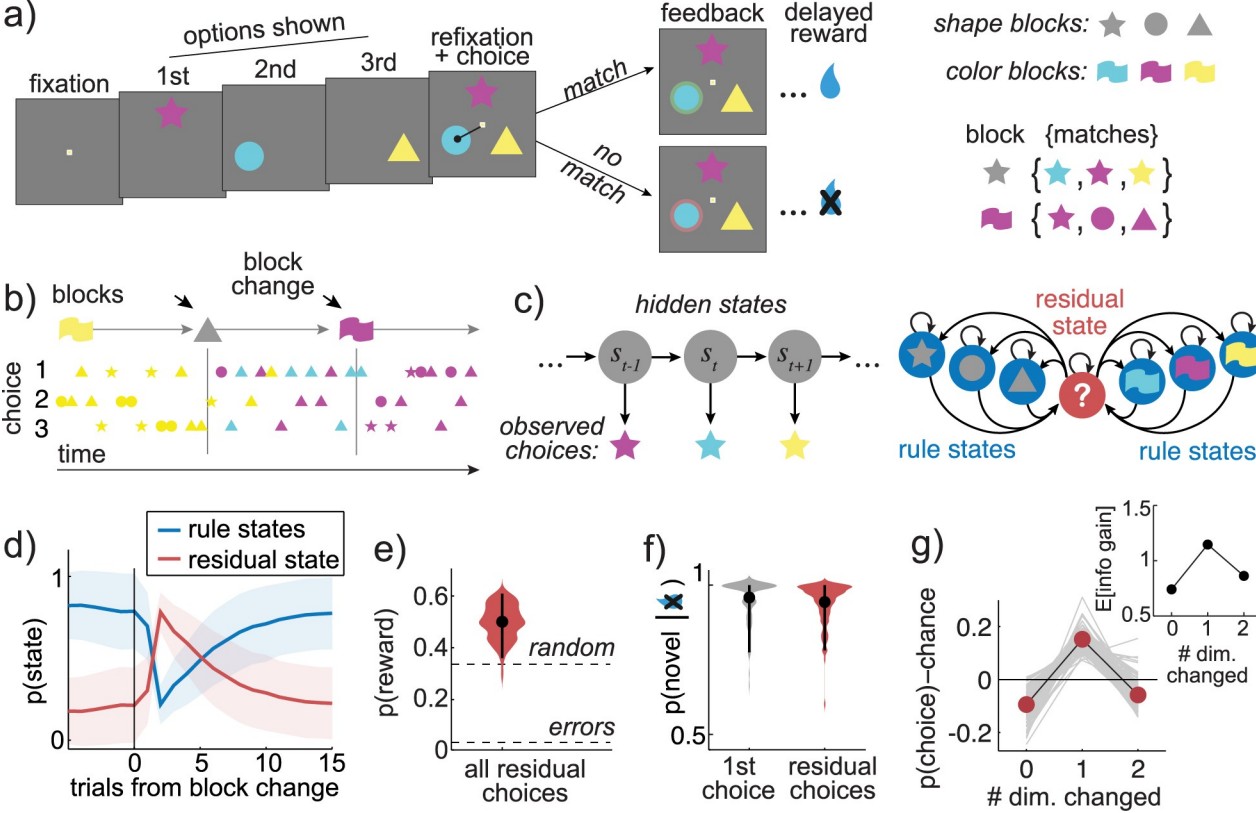

**Fig 1. Task design and latent state identification.** (A) Trial structure in the cognitive set shifting task. Three options are presented sequentially, and the subject indicates choice by refixating a central fixation point, then saccading to 1 option. Visual feedback is displayed, and then the subject is rewarded for choosing an option that matches the current block. (Inset) There are 6 possible blocks, 3 shape blocks (star, circle, and triangle), and 3 color blocks (cyan, magenta, and yellow). Correct choices match the block's rewarded feature but will differ in the irrelevant feature domain. (B) An example sequence of trials, illustrating the blocks (top row). Each symbol reflects the color and shape of 1 choice. (C) Features of the hidden Markov model used to identify rule states. (Left) An unrolled illustration of the model. Choices, illustrated as colored stars, are modeled as emissions from a hidden process that is in one of various latent states at each time point ($s_t$). (Right) The transition paths between latent states in the model. Rule states only emit choices that match the state's feature, while any choice can occur from the residual state. (D) The mean probability of rule-based and residual states, aligned to the block changes. Shaded area = STD across sessions. (E) Accuracy of residual-state choices across sessions (violin plot). Accuracy was greater than expected from errors (bottom dashed line) or random choice (top dashed line). Dot = mean, error bars = 95% CI across sessions. (F) Probability of choosing a novel option (matching neither the last choice's shape or color) after the first omitted reward after a block change (gray) and for all residual-state choices that followed no-reward outcomes (red). Dot = mean, error bars = 95% CI across sessions. (G) The probability of choosing options that differ in 0, 1, or 2 feature dimensions from the last choice on average (red markers) and within individual sessions (gray lines). Compare with the expected information gain about what rule is best for each choice type (inset), calculated from the history of choices and rewards observed in residual-state trials (see also **S2 Fig**). Data: https://doi.org/10.6084/m9.figshare.13139450.v1.

To infer what rule, if any, subjects were applying on each trial, we used a hidden Markov model (HMM; see Methods; **Fig 1C**). Previous rule-based sequential decision-making studies have operationally defined rule-based choices as the set of correct choices within each correct feature block. However, in this task, this approach would (1) include some unambiguously non-rule–based choices (i.e., correct responses made while guessing); and (2) exclude some unambiguously rule-based choices (i.e., perseverative rule-based decisions after the end of the block). We developed an HMM in order to resolve these issues, and the HMM ultimately produced trial labels that better explained variance in neural activity than these traditional approaches (S1 Fig). The HMM modeled each rule as a latent state that produced choices that matched the rule's relevant feature (i.e., the blue rule produced blue choices, regardless of shape). The model allowed us to make statistical inferences about what rule was underlying each choice (i.e., to determine if a blue-star choice was due to a blue rule or a star rule). The rules inferred by the HMM matched the current correct feature 93% of the time on average (±2% standard deviation [STD], 95% CI across sessions: 89.5% to 97.4%). Thus, the model identified the periods when animals were following rules that very often (though not always) matched the rule imposed by the task.

In addition to 6 rule states, the model also included a "**residual**" state to account for all the choices that could not be explained by a simple sensorimotor rule [39]. In the residual state, all choices were modeled as equiprobable, not because they were random, but because this is the maximum entropy distribution for a categorical variable. Using the maximum entropy distribution minimized assumptions about these non-rule–based choices (i.e., about what information was considered, how it was combined, how these computations changed across epochs of the task, etc.). However, it did not preclude the possibility that "residual" choices had some predictable structure, which indeed they did.

## Residual decisions maximized reward and/or information

Residual-state choices occurred more often, but not exclusively, at block changes (**Fig 1D**). On average, across sessions, 45% of residual choices (±21% STD, 95% CI across sessions: 12% to 81%) were within 5 trials of a block change, significantly greater than the expected frequency of 21% (±4%, t(1,127) = 14.8, $p < 0.0001$, paired $t$ test). Conversely, 35% of residual choices were during stable periods (10 or more trials from a change point, ±21% STD, 95% CI across sessions: 5% to 76%), significantly less than the chance frequency of 57% (±8%, t(1,127) = −15.9, $p < 0.0001$, paired $t$ test). This suggests that residual choices increased in frequency when the subjects needed to explore in order to determine what rule to follow. However, residual choices also occurred at times when monkeys could have been following rules but were not: 19.5% of the correct choices that occurred during stable periods (>5 trials after block changes) were not rule-based choices per the HMM (±18% STD, 95% CI across sessions: 1% to 62%). We have previously argued that at least some of these residual choices are due to a tonic exploratory mechanism that causes us to explore both when it is valuable to do so and when it is not [39], but there is controversy about whether choices can be considered exploratory if they occur when exploration has no value [41,42]. The term "residual" thus refers to the entire set of choices that could not be described with a sensory–response rule, including both exploratory choices and other choices. Overall, 67% of trials were classified as rule-based ($n$ = 49,043/73,627), while 33% were classified as residual-state choices ($n$ = 24,584/73,627).

In previous studies, decisions that do not match the current rule are often dismissed as errors—meaning that they are not typically analyzed. However, residual decisions were not necessarily errors here. In fact, the residual decisions identified by the HMM often maximized reward and/or information about what rule was best. For example, residual decisions were

correct 49% of the time (±6.9% STD), regardless of when they occurred in the block. That is, residual choices right after a block change were correct about half the time (47% ± 7% correct, within 5 trials of a block change), and so were residual choices during stable periods (52% ± 13% correct, 10+ trials after). The residual choices were correct far more frequently than we would expect from errors (3% correct) or from random decisions (33% correct; **Fig 1E**). Errors would be correct approximately 3% of the time because this is the probability that an error of rule-following would coincide with a block change and target the new correct feature (significant difference: $p < 0.0001$, t(1,127) = 77, 1-sample $t$ test). Similarly, random decisions would only be correct 33% of the time ($p < 0.0001$, t(1,127) = 26.9, 1-sample $t$ test; both effects survive correction for multiple comparisons). Thus, there is no way to produce choices that are correct 49% of the time through any combination of errors and random decisions. This level of performance is impressive because the largest proportion of residual-state choices occur in the first few trials after block changes, when the animals could not know what the correct feature was. This implies that the monkeys were doing something more complex than random decision-making (like sequential hypothesis testing or avoiding previously unrewarded features), which accelerated their learning during residual-state choices.

Residual choices tended, on average, to maximize information and/or reward. For example, after reward omission, choosing a novel option—one that differs in both color and shape from the last choice—maximizes both reward likelihood and information about the correct feature (**Fig 1H**; see Methods for details on how the information- and reward-maximizing choices were identified). Thus, the optimal choice after reward omission by either criteria is to choose a novel option. On the first trial after feedback that the correct feature has changed, subjects made this optimal choice 95% of the time (±6.7% STD across sessions, sig. different than chance, $p < 0.0001$, t(1,127) = 50.6). Similarly, across all residual choices where the subjects were not rewarded on the last trial, regardless of when these occurred within the block, they made the same novel choice 94% of the time (±6.8%; **Fig 1F**; $p < 0.0001$, t(1,127) = 46.7). Moreover, residual choices in general, regardless of whether they followed reward omission, were strongly biased toward options that maximized information (calculated from the actual pattern of choices and rewards that preceded these choices; **Fig 1G, S2 Fig**; see Methods; probability of changing 1 dimension from the previous choice: 32.6% ± 4.6% STD, paired $t$ test for different than chance at 17.3% ± 2.5% STD, calculated based on how often this option was presented in each session: $p < 0.0001$, t(1,127) = 48.2; mean difference in probability: 34.5% more than chance, 95% CI = 14.6% to 15.2%). Thus, residual choices used a strategy that integrated information about past rewards and choices to maximize information about what rule was best.

## Rule adherence reduces firing rate

We recorded responses of individual neurons in OFC ($n$ = 115 cells), VS ($n$ = 103), and DS ($n$ = 204; recording sites in **Fig 2A**). These regions were chosen because each is implicated in a different aspect of rule-guided decision-making. DS is implicated in rapid, stimulus-bound, automatic decisions [43–45], like the decisions that occur after a rule is well learned. Conversely, VS and OFC are implicated in more flexible, deliberative decision-making [46–50], like the decisions that help us discover rewards or learn new rules in changing environments. Focusing on these regions allowed us to consider an alternative hypothesis about the neurobiological basis of rule-based decision-making: the idea that that rule-based decision-making involves a functional handoff from automatic decision-making structures, like DS, to more flexible decision-making structures, like OFC and/or VS [21–24]. We will refer to this idea as the "handoff hypothesis" to differentiate it from the idea that rules are implemented via warping the distributed representation of choice information across regions.

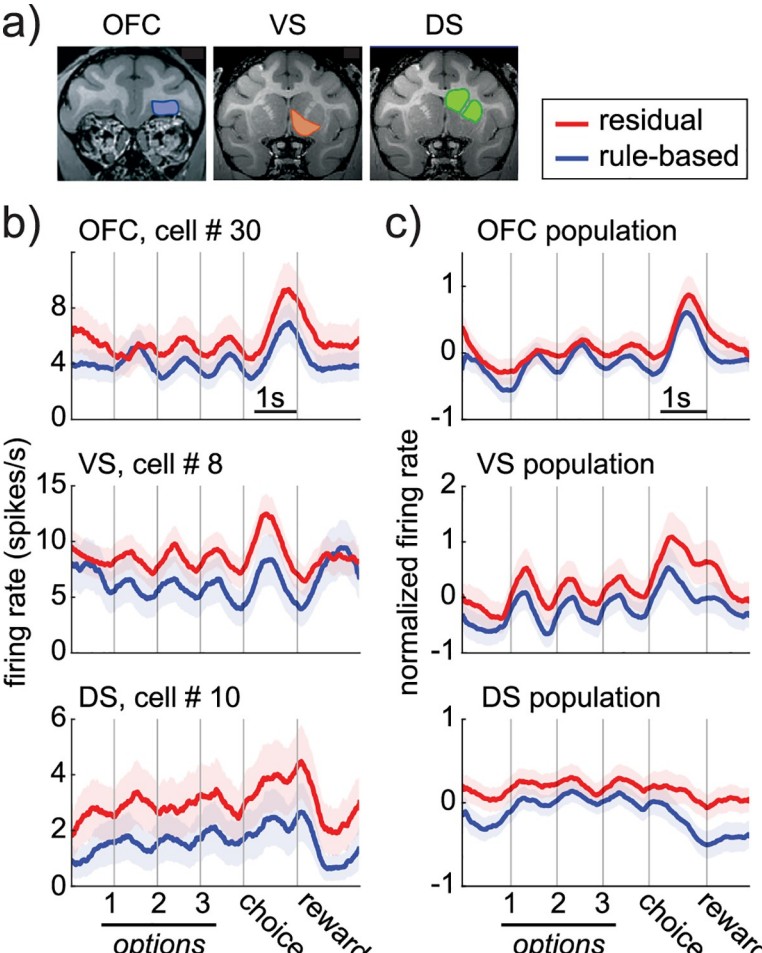

**Fig 2. Firing rate decreases during rule-based decisions.** (A) Coronal slices illustrating the approximate center of recording sites in the OFC, VS, and DS. See Methods for detail of the spatial extent of recording sites. (B) Example cells in the OFC (top), VS (middle), and DS (bottom), recorded during rule-based (blue) and residual (red) decisions. Shaded areas = standard errors across trials. (C) Same as B for the populations. Shaded areas = standard errors across neurons. Data: https://doi.org/10.6084/m9.figshare.13139450.v1. DS, dorsal striatum; OFC, orbitofrontal cortex; VS, ventral striatum.

We first looked for differences in overall spiking activity between rule-based decisions and residual decisions. If rule-based decisions are more metabolically and computationally efficient than other choices, fewer spikes should be required to generate them across the distributed network of decision-making regions. However, the alternative, handoff hypothesis might predict a different pattern—like heterogeneous shifts in the concentration of neural activity across structures. Across the option viewing and choice period (before feedback), we found that firing rates were systematically lower during rule-based decisions in all 3 regions (example cells: **Fig 2B**; population: **Fig 2C**; VS: reduction of 0.38 spikes/s ± 0.14 standard error [STE] across neurons, $p < 0.0001$; DS, reduction of 0.19 ± 0.10 spikes/s, $p < 0.0001$; OFC, reduction of 0.17 ± 0.11 spikes/s, $p < 0.0001$; permutation test against expected difference with shuffled state labels; all effects survive correction for multiple comparisons). A large proportion of individual cells differentiated between rule-based and residual choices in each region (**Fig 3A**; 1-way ANOVAs within each neuron; VS: $n = 45/103$, proportion = 0.44, $p < 0.0001$; DS: $n = 72/204$; proportion = 0.35, $p < 0.0001$; OFC: $n = 49/115$, proportion = 0.43, $p < 0.0001$;

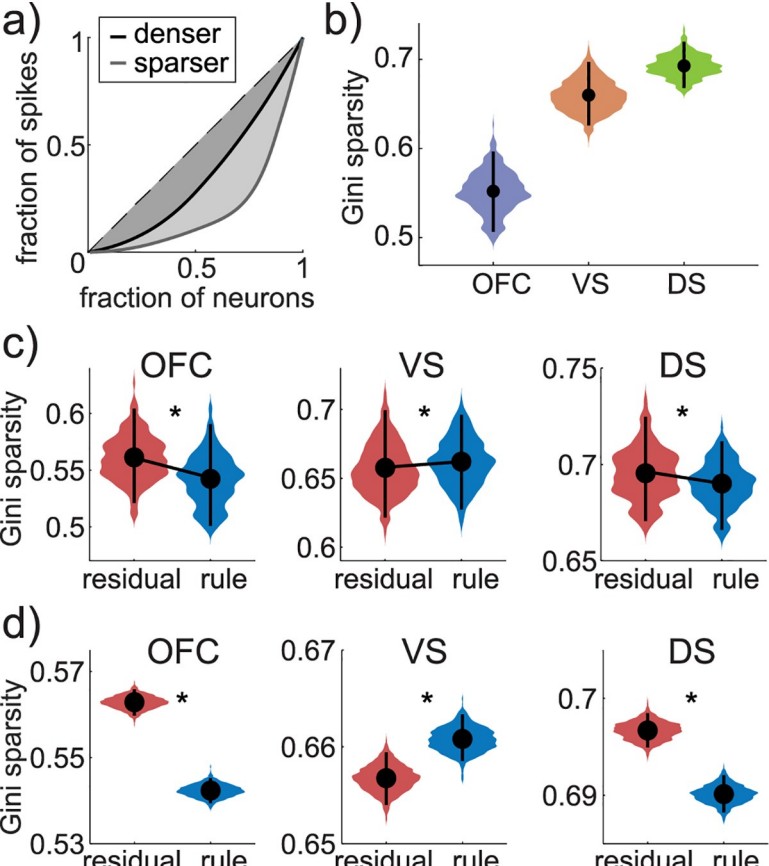

**Fig 3. Heterogeneous changes in the sparsity of neural activity across regions.** (A) The Gini index is a measure of the sparsity of the distribution of spikes across a neuronal population. As more spikes are concentrated in a smaller number of neurons (light trace), the area under the curve (shaded region) increases, increasing the Gini index. (B) The distribution of Gini indices across pseudotrials in 1 example pseudopopulation, across OFC (purple), VS (orange), and DS (green). Dots = mean across pseudotrials, bars = 95% CI. (C) Distribution of Gini indices plotted separately for rule-based (blue) and residual (red) pseudotrials in OFC (left), VS (middle), and DS (right). Dots = mean across pseudotrials, bars = 95% CI. Asterisks = $p < 0.02$, 2-sample $t$ test. (D) Distribution of the mean Gini indices for rule-based and residual trials across 1,000 pseudopopulations. Dots = mean across pseudopopulations, bars = 95% CI. Asterisks: $p < 0.05$, paired bootstrap test. Data: https://doi.org/10.6084/m9.figshare.13139450.v1. DS, dorsal striatum; OFC, orbitofrontal cortex; VS, ventral striatum.

1-sided binomial test for difference from expected false positive rate of 0.05). The decrease in activity during rule-based decisions was not due to differences in reward expectation between the 2 conditions because reward-dependent changes in firing rate were both smaller and in the opposite direction in all 3 regions (no reward–reward, VS: $-0.28 \pm 0.09$, DS: $-0.09 \pm 0.07$, OFC: $-0.06 \pm 0.07$). Thus, if anything, we would be underestimating the extent to which adhering to a rule decreases spiking activity in these regions because rule-based choices were more likely to be rewarded than residual choices.

## Rule adherence has different effects on sparsity across regions

The fact that we observed similar firing rate effects in all 3 regions differs from what we may have expected from functional magnetic resonance imaging (fMRI) studies. fMRI studies [47,49,51] tend to report the kinds of complex shifts in activity between regions that we would expect from the handoff hypothesis: a dissociation between VS and OFC on one hand (where

the blood-oxygen-level-dependent (BOLD) signal is strongest during flexible decision-making and learning) and DS on the other (where the BOLD signal is strongest during rapid, stimulus-bound, automatic decisions, like the rule-based decisions here). This apparent discrepancy could reflect the fact that the BOLD signal may not scale with the rate of spikes, but instead with other population measures, such as how densely or sparsely activity is distributed across a population of neurons [52]. Therefore, we next asked whether there were dissociable changes in the sparsity of activity across VS, DS, and OFC during rule-based and residual-state decision-making (**Fig 3**; see Methods; [53]).

In VS, the population response was more sparse during rule-based decisions (Gini index = 0.662) than residual decisions (0.658, 2-sample *t* test, $p < 0.02$, t(1,358) = 2.35, 95% CI for the difference = [0.0007, 0.0082]; mean difference across pseudopopulations = −0.004, 95% CI = [−0.008, −0.0003], $p < 0.03$, paired bootstrap test). However, there was an opposite pattern in both DS (rule-based = 0.690, residual = 0.696, $p < 0.0001$, t(1,358) = −4.69, 95% CI = [−0.009, −0.004]; mean difference across pseudopopulations = 0.007, 95% CI = [0.004, 0.009], $p < 0.001$, paired bootstrap test) and OFC (rule-based = 0.541, residual = 0.562, $p < 0.0001$, t(1,358) = −9.54, 95% CI = [−0.03, −0.02]; mean difference across pseudopopulations = 0.02, 95% CI = [0.016, 0.025], $p < 0.001$, paired bootstrap test; all effects survive correction for multiple comparisons). Thus, while spiking was lower during rule-based decisions in all 3 regions, spikes were more densely distributed in VS during residual-state decision-making, but more densely distributed in OFC and DS during rule-based decision-making. This suggests that differences between **Fig 2** and results of related work in fMRI [47,49,51] may be due to differences in what is measured by the BOLD signal versus single-unit electrophysiology, though we cannot rule out alternative explanations, like differences between tasks, species, data analysis methods, or limitations in our spatial sampling within OFC, VS, and DS. Together, the firing rate and sparsity effects imply that fMRI evidence that supports a handoff in control may be due to inter-structure heterogeneity in how rules affect the density of spikes across neuronal populations, rather than heterogeneity in how rules affect net neural activity. Instead, across a distributed network of decision-making regions, we found that fewer spikes were needed to generate rule-based decisions.

## Rule adherence increases information about choice identity while reducing metabolic costs

Spikes are energetically costly [13,14], so a distributed decrease in neural activity during rule-based decisions could imply a distributed decrease in the energetic costs of rule-based decisions. This is precisely what we would expect if the brain takes advantage of the opportunity rules present to streamline decision-making. However, a more efficient code requires both (1) that fewer spikes be expended (as we have seen); and (2) that information not be lost, despite this decrease in firing rate. Alternatively, a decrease in spike rates could suggest that a structure is disengaged—perhaps fewer spikes are fired during rule-based decisions because less information is encoded in that structure. Further, if the handoff hypothesis is correct, the distributed decrease in net activity across OFC, VS, and DS could be due to different effects within each structure. Specifically, during rule-based decisions, net activity may go down in OFC and VS because the regions are disengaged (in which case choice information should also go down), whereas net activity may only go down in DS because there is less irrelevant information coming in from OFC and VS (in which case choice information in DS should go up). To differentiate between these possibilities, we next asked whether neurons in these regions encoded choice information and whether the energetic (spiking) cost of encoding this information was altered during rule-based decisions.

We found that neurons in all 3 areas encoded the visual features of the upcoming choice ("choice identity;" **Fig 4A–4C;** 1-way ANOVAs, predicting choice identity from the firing rate of each neuron; VS: $n = 52/103$, proportion = 0.50, $p < 0.0001$; DS: $n = 85/204$; proportion = 0.41,

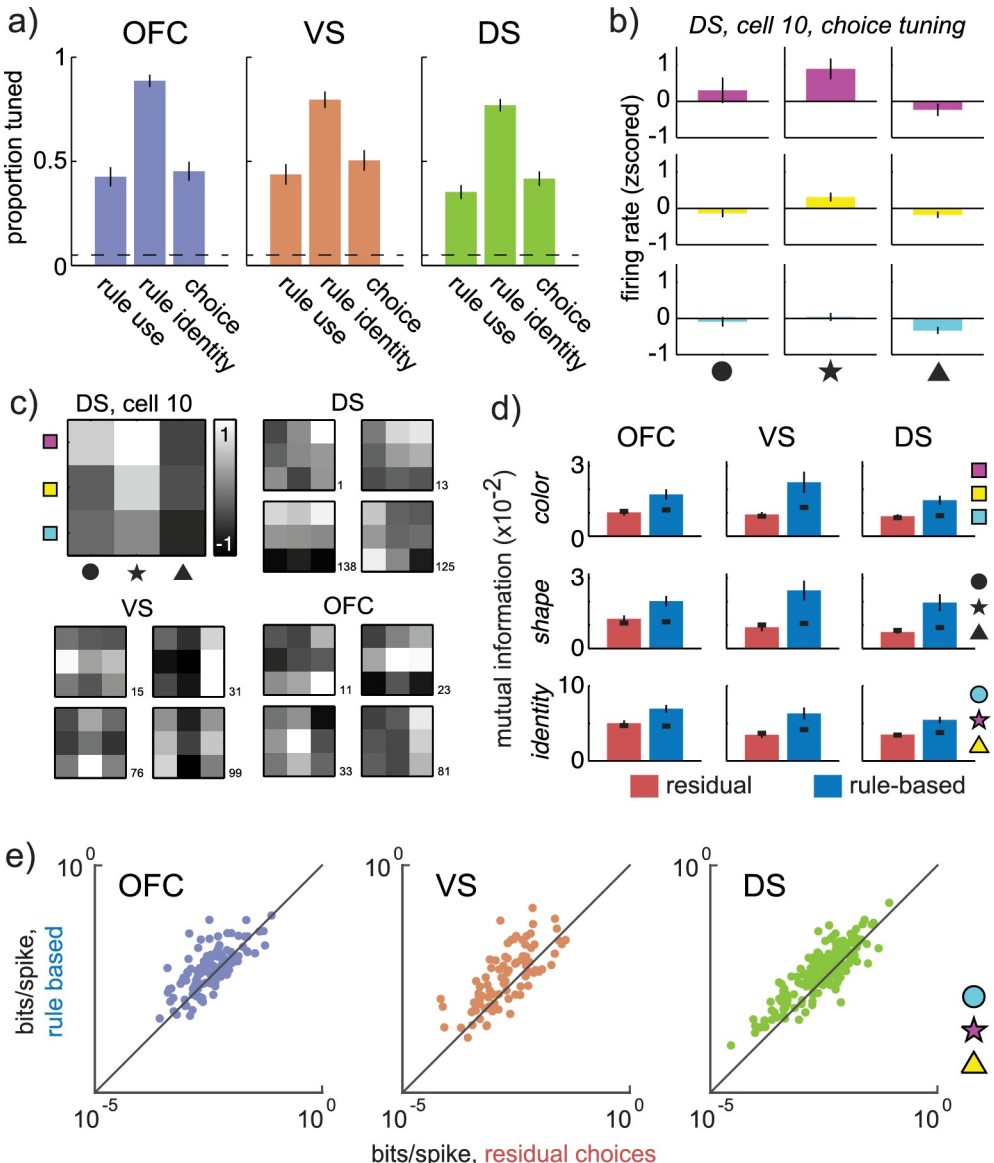

**Fig 4. Choice information increases during rule-based decisions.** (A) The proportion of single neurons in each region that were tuned for rule use (differentiated rule-based from residual choices), rule identity (the color or shape rule underlying rule-based choices), and choice identity (the color and shape of the chosen option). Error bars = binomial standard error, dotted line = proportion expected by chance. (B) Mean firing rate of an example DS neuron for each choice identity. Error bars = STE across trials. (C) The same example neuron represented as a choice tuning map (top left), and the tuning maps of 4 additional example neurons from DS, VS, and OFC. Shading represents z-scored firing rate, and the color bar is shared across plots. (D) Information about the color (top row), shape (middle row), and identity (bottom row) of the chosen option in each of OFC (left column), VS (middle column), and DS (right column). Error bars = STE across neurons and thick black lines = shuffled data. (E) Choice identity information per spike during rule-based (y-axis) and residual decisions (x-axis), measured in individual cells (dots) recorded in each of OFC (left, purple), VS (middle, orange), and DS (right, green). Dots on the unity line indicate no change in bits/spike across rule-based and residual decisions; dots above unity indicate an increase in bits/spike during rule-based decisions. Log scale. Data: https://doi.org/10.6084/m9.figshare.13139450.v1. DS, dorsal striatum; OFC, orbitofrontal cortex; STE, standard error; VS, ventral striatum.

$p < 0.0001$; OFC: $n = 52/115$, proportion = 0.45, $p < 0.0001$; 1-sided binomial test for difference from expected false positive rate of 0.05). To determine how the amount of choice information changed across rule-based decisions and residual ones, we calculated the mutual information between each neuron's firing rate and the chosen option separately for each choice type (see Methods). Choice-predictive information was increased during rule-based decisions, compared to residual decisions, in all 3 regions: VS (rule-based: average of 0.06 bits per trial ± 0.008 SEM, residual: 0.03 ± 0.004, $p < 0.0001$, z = 4.40, $n = 103$, Wilcoxon rank sum test), DS (rule-based: 0.05 ± 0.004, residual: 0.03 ± 0.003, $p < 0.0001$, z = 6.21, $n = 204$), and OFC (rule-based: 0.07 ± 0.005, residual: 0.05 ± 0.004, $p < 0.0002$, z = 3.77, $n = 115$; all effects survive correction for multiple comparisons). Although mutual information was lower during residual decisions, some neurons did encode choice identity during residual decisions according to other metrics (1-way ANOVAs, predicting chosen stimulus identity from the firing rate of each neuron; VS: $n = 9/103$, proportion = 0.09, $p < 0.04$; DS: $n = 23/204$, proportion = 0.11, $p < 0.0001$; OFC: $n = 14/115$, proportion = 0.12, $p < 0.001$; 1-sided binomial test for difference from expected false positive rate of 0.05). Thus, choice information increased during rule-based decisions but was not completely absent during residual decisions.

The increase in choice information during rule-based decisions was not a trivial consequence of the difference in firing rate between the 2 types of decisions. First, in the absence of any changes in tuning functions, we would expect mutual information to increase with mean firing rate because increasing spike counts moves activity away from the noise floor [54,55]. However, here, firing rates were lower during rule-based decisions than residual ones. Thus, if anything, we are likely underestimating the magnitude of the increase in choice information during rule-based decisions. Second, the information per spike also increased during rule-based decisions in all 3 regions (VS, rule-based: 0.012 bits per spike ± 0.005, residual: 0.004 ± 0.001, $p < 0.0002$, z = 3.84, DS, rule-based: 0.008 ± 0.001, residual: 0.005 ± 0.0006, $p < 0.0001$, z = 6.11, and OFC, rule-based: 0.009 ± 0.001, residual: 0.007 ± 0.001, $p < 0.002$, z = 3.11; all effects survive correction for multiple comparisons). Because spikes are metabolically costly [13,14], the fact that more information was conveyed per spike in every region we tested suggests that rule-based decisions may be more energetically efficient than the other decisions. Fewer spikes were wasted on computations that were irrelevant to the ultimate choice. This result is broadly consistent with the idea that rules can streamline decision-making, perhaps via distributed changes in the way that neurons are tuned for or representing choice information. We will look for these changes next. However, this result is not consistent with the predictions of the handoff hypothesis—where we would have expected an increase in information per spike in DS (because this region would be driving rule-based decisions), but a decrease in information per spike in VS and OFC (because these regions would have handed control off to DS).

## Rule adherence warps choice representations

Together, these results broadly support the idea that rule-based decisions make more efficient use of neural resources than the other decisions in this task. However, our hypothesis is that rule-based decisions should be more efficient than other decisions because of changes in how choices are represented in decision circuits. Though we cannot directly ask how different goal states change cognitive representations [56], we can take advantage of the fact that neural representations have metric relationships to each other that recapitulate cognitive similarities [57,58]. This means that we can ask how the similarity of neural representations of different choices changes under different goal states (see Methods; [59]). Here, we define a choice representation within decision-making regions as a pattern of firing rates across a population of neurons, or equivalently, as vector in neuron-dimensional state space (**Fig 5A**). Because

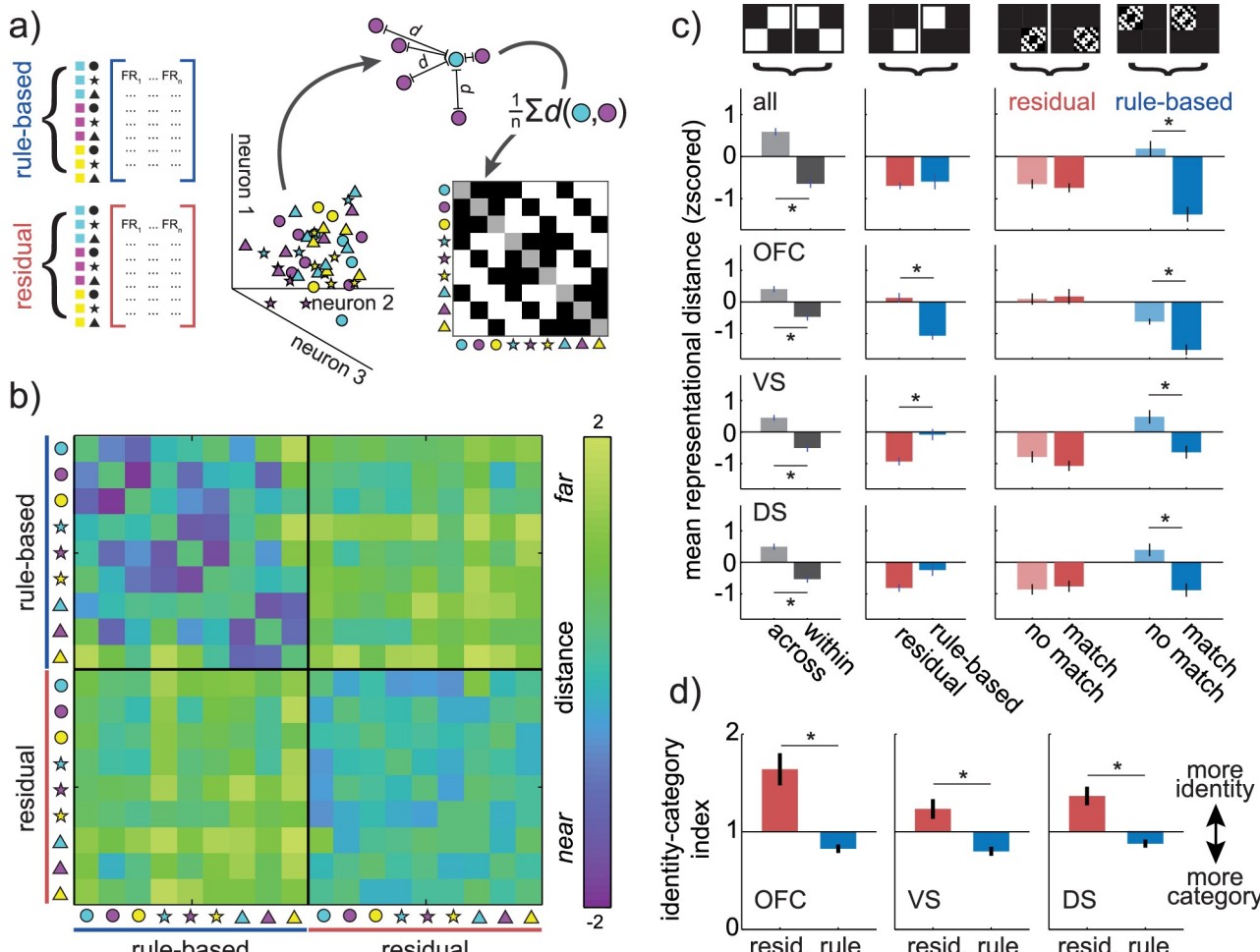

**Fig 5. Changes in choice representations across rule-based and residual-state decision-making. (A)** Separately recorded neurons were combined into rule-based and residual pseudopopulations (left). This allowed us to estimate the position of different types of pseudotrials in neuron-dimensional state space (middle). To determine how different task conditions affected the geometry of neural representations, the average distance between different types of choices (here, between cyan circle and magenta circles) is calculated and used to populate a representational similarity matrix (right). The example matrix shows the structure this matrix will have if choices that share a color or shape are represented by more similar patterns of activity across neurons than choices that do not share features (i.e., if these choices are closer together in neuronal state space). **(B)** The representational similarity matrix for both rule-based (upper left) and residual (bottom right) decisions, calculated across all regions, as well as the similarity between rule-based and residual representations (off-diagonal blocks). **(C)** Specific contrasts across the representational similarity matrix for all regions, then separately for OFC (top row), VS (middle), and DS (bottom). (Left column) If there is a change in how choices are represented between rule-based and residual decisions, then the mean distance between rule-based and residual choice representations (light gray, average of off-diagonal blocks) should be greater than the distance within decision types (dark gray, average of on-diagonal blocks). (Middle column) If there is a change in the total representational space between decision types, then the mean distance within rule-based decisions (blue, average of top left block) should differ from residual decisions (red, average of bottom right block). (Right column) If the categorical structure of the choices is represented, then the mean distance between stimuli that share features ("match," dark color) should be smaller than the distance between choices that do not share features ("no match," light color). This is calculated separately for both residual (red) and rule-based (blue) decisions. Error bars = STE across unique matrix elements, asterisks = significant contrasts in both the example pseudopopulation (all $p < 0.001$, paired $t$ test) and across pseudopopulations (all $p < 0.001$, bootstrap test, **S2 Fig**). **(D)** The identity–category index for single neurons in OFC (left), VS (middle), and DS (left). Error bars = STE across neurons, asterisks = significant contrasts. Data: https://doi.org/10.6084/m9.figshare.13139450.v1. DS, dorsal striatum; FR, firing rate; OFC, orbitofrontal cortex; STE, standard error; VS, ventral striatum.

neurons were recorded largely separately, we built representational vectors from pseudopopulations (see Methods; [22,60–62]).

First, we asked whether there were representational changes due to rule-based decision-making in general, rather than changes due to applying a specific rule. Because we had no

knowledge of which, if any, stimulus feature was focal during the residual decisions, the residual-state pseudopopulation necessarily combined trials in which different stimulus features were focal. To create an equivalent rule-based pseudopopulation, rule-based trials were sampled without regard for the rule. We first combined the data across VS, DS, and OFC, asking how the distances between distributed choice representations changed across choices and goals [59]. We illustrate the pairwise distance between the representations of all possible choices during both rule-based and residual-state decision-making as a representational similarity matrix (**Fig 5B**; see Methods).

There was a representational shift between rule-based and residual decisions (ANOVA, main effect of within/between choice type: $F_{(1,147)} = 138.9$, $p < 0.0001$; main effect of rule-based/residual choice type: $F_{(1,147)} = 5.0$, $p < 0.03$; main effect of matching features: $F_{(1,147)} = 23.3$, $p < 0.0001$; interaction between within/between and matching features: $F_{(1,147)} = 7.9$, $p < 0.006$; interaction between rule-based/residual and matching features: $F_{(1,147)} = 32.0$, $p < 0.0001$). This representational shift was due to 3 changes in the representational similarity matrix (**Fig 5C**, **S3 Fig**). First, choice representations changed across rule state. Choice representations were closer to themselves within choice type than between choice types (within: mean distance = $-0.64 \pm 0.10$ STE, between = $0.59 \pm 0.09$, $p < 0.0001$, $t_{(1,71)} = 9.62$, 95% CI = [0.98, 1.49]). However, second, this was not due to a simple expansion of the representational space during rule-based decisions because the average distance between choice representations was unchanged across choice types (rule-based: mean distance = $-0.59 \pm 0.18$, residual: mean distance = $-0.69 \pm 0.08$, $p > 0.5$, $t_{(1,35)} = 0.57$, 95% CI for effect size = [$-0.26$, 0.47], paired $t$ test). Third, choices that shared a feature were closer than choices that did not share a feature only during rule-based decision-making (shared feature = $-1.37 \pm 0.18$; no shared feature = $0.18 \pm 0.18$, $p < 0.0001$, $t_{(1,34)} = 6.12$, 95% CI = [1.03, 2.06], 2-sample $t$ test), not during residual-state decision-making (shared feature = $-0.74 \pm 0.10$; no shared feature = $-0.65 \pm 0.11$, $p > 0.5$, $t_{(1,34)} = 0.58$, 2-sample $t$ test). Thus, choice representations shifted such that similar choices were represented by more similar patterns of activity during rule-based decisions. As a result, choice representations become more organized with respect to stimulus categories during rule-based decisions.

We next asked whether these effects were the same across OFC, VS, or DS or driven by representational changes in one of the regions. Choices were more similar within rule-based or residual-state decisions than between choice types in all 3 regions, indicating that representational shifts did occur in each region (OFC: within = $-0.47 \pm 0.12$ STE, between = $0.41 \pm 0.10$, $p < 0.0001$, $t_{(1,71)} = 7.16$; VS: within = $-0.50 \pm 0.12$, between = $0.45 \pm 0.09$, $p < 0.0001$, $t_{(1,71)} = 6.39$; DS: within = $-0.53 \pm 0.11$, between = $0.49 \pm 0.09$, $p < 0.0001$, $t_{(1,71)} = 7.74$; bootstrapped population results for this and all following analyses related to **Fig 5** are in **S3 Fig** and **Table A in S1 Text**; cross-validated results in **S4 Fig**).

The regions did differ in how the total size of the representational space changed across choice types. In OFC, choice representations were closer together, on average, during rule-based decisions than residual decisions (rule-based = $-1.07 \pm 0.12$, residual = $0.13 \pm 0.15$, $p < 0.0001$, $t_{(1,35)} = 6.48$, paired $t$ test). In VS, the representational space was expanded during rule-based decision-making, compared to residual-state decision-making (VS: rule-based = $-0.08 \pm 0.18$, residual = $-0.93 \pm 0.12$, $p < 0.0001$, $t_{(1,35)} = 5.17$). In DS, it was expanded in the 1 example population (rule-based = $-0.25 \pm 0.18$, residual = $-0.81 \pm 0.12$, $p < 0.02$, $t_{(1,71)} = 2.60$), but this effect did not replicate across 1,000 bootstrapped repetitions.

Nevertheless, in all 3 regions, choices that shared a feature were only closer to each other than choices that did not share a feature during rule-based decisions (OFC: shared feature = $-1.52 \pm 0.17$, no shared feature = $-0.62 \pm 0.09$, $p < 0.0001$, $t_{(1,34)} = 4.63$, 2-sample $t$ test; VS: shared feature = $-0.64 \pm 0.20$, no shared feature = $0.48 \pm 0.21$, $p < 0.001$, $t_{(1,34)} = 3.82$; DS:

shared feature = −0.88 ± 0.21, no shared feature = 0.39 ± 0.20, $p < 0.0002$, t(1,34) = 4.40; all effects survive correction for multiple comparisons within the example population and across populations). During residual-state decisions, the choices that shared a feature were represented no more similarly than choices that did not (OFC: shared feature = 0.17 ± 0.24, no shared feature = 0.09 ± 0.18, $p > 0.7$, t(1,34) = 0.27, 2-sample $t$ test; VS: shared feature = −1.07 ± 0.16, no shared feature = −0.79 ± 0.18, $p > 0.2$, t(1,34) = 1.19; DS: shared feature = −0.77 ± 0.18; no shared feature = −0.86 ± 0.16; $p > 0.7$, t(1,34) = 0.38). Thus, while choice representations were only more compact during rule-based decision-making in OFC—not in striatum—choice representations shifted so that they were more categorically organized during rule-based decision-making in all 3 regions. This is broadly consistent with our overarching warping hypothesis because reducing the encoding of rule-irrelevant information would make decisions that differ only in rule-irrelevant ways more similar to each other—decreasing the distance between representations of choices that share a category in neuronal state space.

## Single-neuron choice tuning emphasizes category, rather than identity, during rule-based decisions

Because the choice representations were constructed from pseudopopulations rather than simultaneously recorded populations, the increase in category encoding during rule-based decisions could have been an artifact of how we combined activity across neurons. However, representational changes at the population level should also be associated with changes in the tuning of single neurons [59]. Here, the changes in choice representations at the population level suggested that single neurons should have less tuning for choice identity (combination of color and shape), but more tuning for choice category (color or shape) during rule-based decisions.

We quantified the relative amount of choice identity and choice category tuning in single neurons with an "identity–category index" (see Methods). We found a significant decrease in the identity–category index (relative increase in categorization) during rule-based in all 3 regions (**Fig 5D**): in neurons recorded from OFC (residual = 1.64 ± 0.16 STE, rule-based = 0.83 ± 0.04, sig. decrease, $p < 0.0001$, t(1,114) = −4.77, paired $t$ test), VS (residual = 1.23 ± 0.10, rule-based = 0.80 ± 0.05, $p < 0.0002$, t(1,102) = −3.95), and DS (residual = 1.37 ± 0.09 STE, rule-based = 0.88 ± 0.04, $p < 0.0001$, t(1,202) = −4.59; all effects survive correction for multiple comparisons). Thus, during rule-based decisions, even single neurons shifted from encoding choice identity to encoding choice category, suggesting that the emergence of categorical structure at the population level was not an artifact.

## Color rules and shape rules have different effects on choice representations

Population choice representations and single-neuron tuning functions both became more categorically organized during rule-based decision-making. One way to introduce categorical structure—to make choice representations that share a feature more similar to each other—is to eliminate the encoding of the rule-irrelevant information that would otherwise pull them apart. This would be consistent with our overarching view that rule-based decisions are an efficient use of neural resources because we take advantage of the opportunity to eliminate the encoding of rule-irrelevant information. However, if this is the reason for the categorical structure, then representations of choices that share a color should only come together when color is irrelevant—during shape rules—while representations that share a shape should only come together when shape is irrelevant—during color rules.

To test this idea, we next performed similar analyses on pseudopopulations constructed separately for shape and color rules (**Fig 6A**). Combining across OFC, VS, and DS, we saw

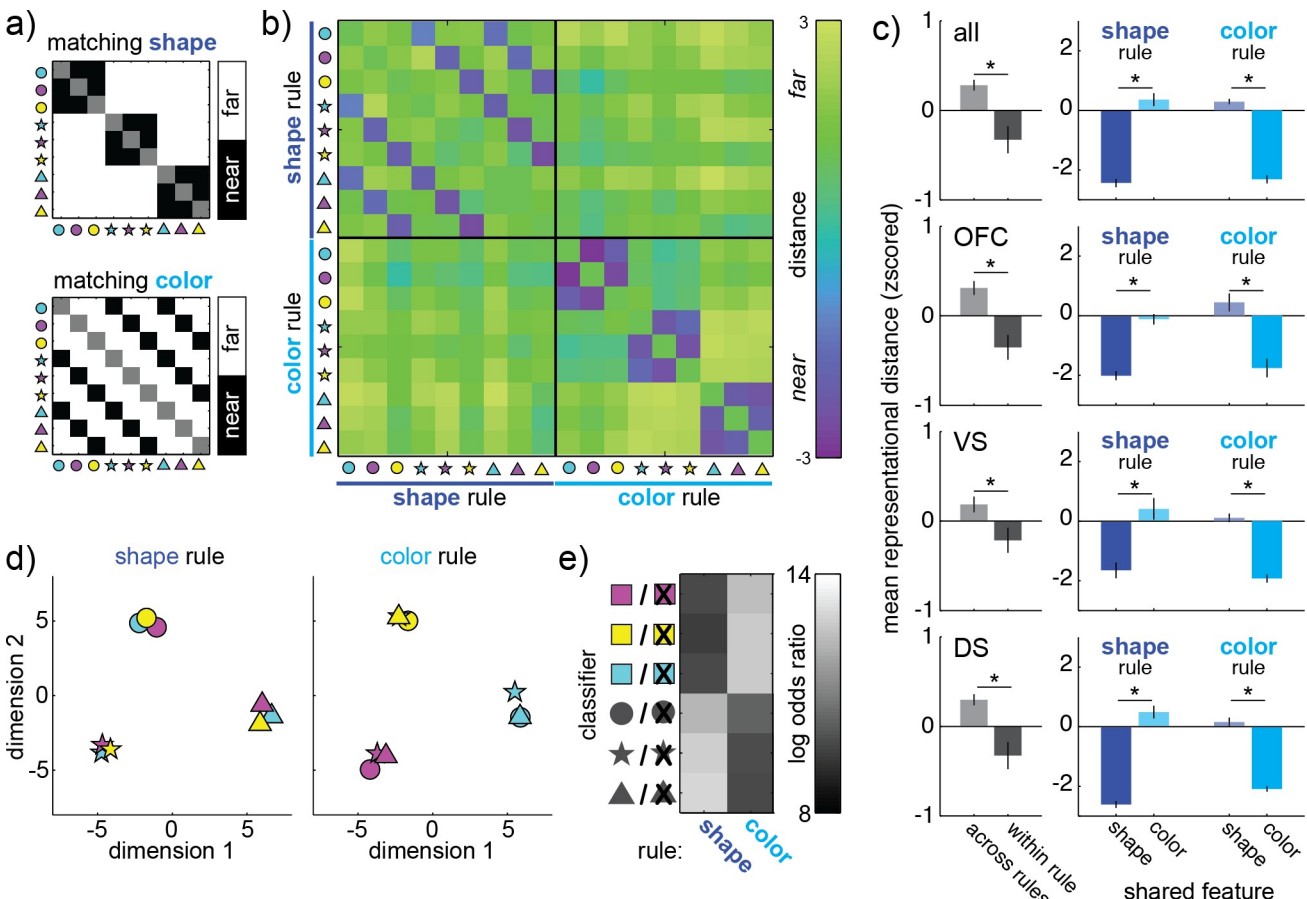

**Fig 6. Changes in choice representations during color and shape rules.** (A) (Top) To contrast shape-rule and color-rule choice representations, separate pseudopopulations are constructed for each class of rule. (Bottom) The expected outcome if only shape (left) or color (right) influenced representational similarity. (B) The representational similarity matrix for both shape-rule (upper left) and color-rule (bottom right) decisions, all regions. (C) Contrasts across the representational similarity matrix, calculated for all regions (top), then separately for OFC, VS, and DS (top to bottom). (Left column) A change in representation between color-rule and shape-rule decisions would increase distance between representations measured across decision types (light gray), compared to within decision types (dark gray). (Right column) Mean distance between stimuli that share the rule-relevant feature (saturated color) and rule-irrelevant feature (dim color), during shape-rule (left column) and color-rule (right column) decisions. Error bars = STE across pairs of trial types. (D) Multidimensional scaling plots illustrate how choice representations shift across shape-rule (left) and color-rule (right) decisions. Dimensions calculated separately for shape- and color-rule trials. (E) The log odds ratio (chosen−unchosen) for each choice category classifier (y-axis) across shape and color rules (x-axis). Classifiers trained and fit to the pseudotrial matrix. Lighter shades = more confidence in the correct classification. Data: https://doi.org/10.6084/m9.figshare.13139450.v1. DS, dorsal striatum; OFC, orbitofrontal cortex; STE, standard error; VS, ventral striatum.

significant change in the organization of neural representations across rule types (ANOVA, main effect of within/between rule type: $F(1,135) = 131.1$, $p < 0.0001$; main effect of rule type: $F(1,135) = 0.12$, $p > 0.7$; main effect of matching features: $F(1,135) = 43.6$, $p < 0.0001$; interaction between within/between and matching features: $F(1,135) = 44.4$, $p < 0.0001$; interaction between rule-type and matching features: $F(1,135) = 172.4$, $p < 0.0001$). This was due to a variety of effects. First, a representational shift between color-rule and shape-rule choice representations (mean distance within rule type = $-0.33 \pm 0.15$ STE, between rule types = $0.28 \pm 0.06$; $p < 0.002$, $t(1,71) = 3.41$, 95% CI = [0.25, 0.96]). However, this representational shift specifically decreased the representational distance between rule-irrelevant features, both during the color rule (mean distance between same-color choices = $0.35 \pm 0.22$, same-shape = $-2.43 \pm 0.14$, $p < 0.0001$, $t(1,16) = 10.70$, 95% CI = [2.24, 3.35]), and during the shape rule (mean distance

between same-shape choices = 0.29 ± 0.09, same-color = −2.31 ± 0.14, $p < 0.0001$, t(1,16) = 15.62, 95% CI = [2.25, 2.96]). Thus, there was an increase in the representational similarity of choices that differed solely along rule-irrelevant dimensions, consistent with the predictions of our overarching hypothesis. However, other interpretations are possible. For example, choices that share both a rule domain and a feature tend to be more tightly clustered in time than choices that share a feature, but not a rule domain. Temporal autocorrelations could also drive these similarity effects. Further, choices that share both a rule domain and a feature necessarily also share a specific rule—blue is the only color rule that would lead to blue-star, blue-circle, or blue-triangle choices. Any abstract encoding of a blue rule would also drive the representation of these choices to appear more similar.

We observed similar representational changes within each region individually: OFC (within rule types = −0.35 ± 0.14, between = 0.31 ± 0.08, $p < 0.0002$, t(1,71) = 4.01, 95% CI = [0.33, 0.99]), VS (within rule types = −0.21 ± 0.14, between = 0.19 ± 0.09, $p < 0.03$, t(1,71) = 2.27, 95% CI = [0.05, 0.75]), and DS (within rule types = −0.32 ± 0.15, between = 0.30 ± 0.06, $p < 0.0005$, t(1,71) = 3.72, 95% CI = [0.28, 0.96]). Within each region, the representational shifts were identical to what we observed when combining across regions (**Fig 6**, **S5 Fig**, and **Table B and C in S1 Text**; all effects survive correction for multiple comparisons within the example population and across populations). Thus, we saw the same rule-selective representational changes in each region. This could suggest that the categorical structure we observed in **Fig 5** emerged because similar choices that differed only in rule-irrelevant ways were represented more similarly, though, again, other interpretations are possible, such effects due to the temporal structure of the task or abstract encoding of the current rule.

## Rule-relevant coding dimensions expand and rule-irrelevant dimensions compress during rule-based decision-making

Although the results are broadly consistent with our overarching hypothesis, an increase in similarity could also be a trivial artifact of an abstract rule identity signal. Simply encoding a blue rule during blue star, circle, and triangle choices could make the representation of these choices more similar, regardless of shape. Indeed, this is a strong possibility here: slightly more than 80% of single neurons encoded rule identity (**Fig 4A**; 1-way ANOVAs within each neuron, excluding residual choices; VS: $n = 82/103$, proportion = 0.80, $p < 0.0001$; DS: $n = 157/204$; proportion = 0.77, $p < 0.0001$; OFC: $n = 102/115$, proportion = 0.89, $p < 0.0001$; 1-sided binomial test for difference from expected false positive rate of 0.05). Therefore, we next asked whether these representational changes were due to a meaningful change in the way that choices were computed, rather than a trivial effect of rule encoding.

Subjects could make the same choice for 3 different reasons, which occurred with approximately equal frequencies. They could make a blue-star choice because they were following a blue rule, following a star rule, or making residual-state decisions (**Fig 7A**). This meant that it was possible to identify the patterns of neural activity that predicted blue choices, regardless of the goal, by marginalizing across goals, then to examine how goals affected the encoding of choice information. To identify these patterns, we used targeted dimensionality reduction to identify choice-predictive coding dimensions in neuronal state space (**Fig 7B**; see Methods). These are the linear combinations of neuronal firing rates (equivalently, directions in neuron-dimensional space) along which activity predicts the choice the monkey will actually make. We identified the 3 coding dimensions that predicted choice color and 3 coding dimensions that predicted choice shape. As expected, choice-predictive coding dimensions were neither identical nor orthogonal to the matching rule coding dimensions (mean angle between rule and matching choice coding dimensions = 52.7 degrees, range across choice categories = [48.5,

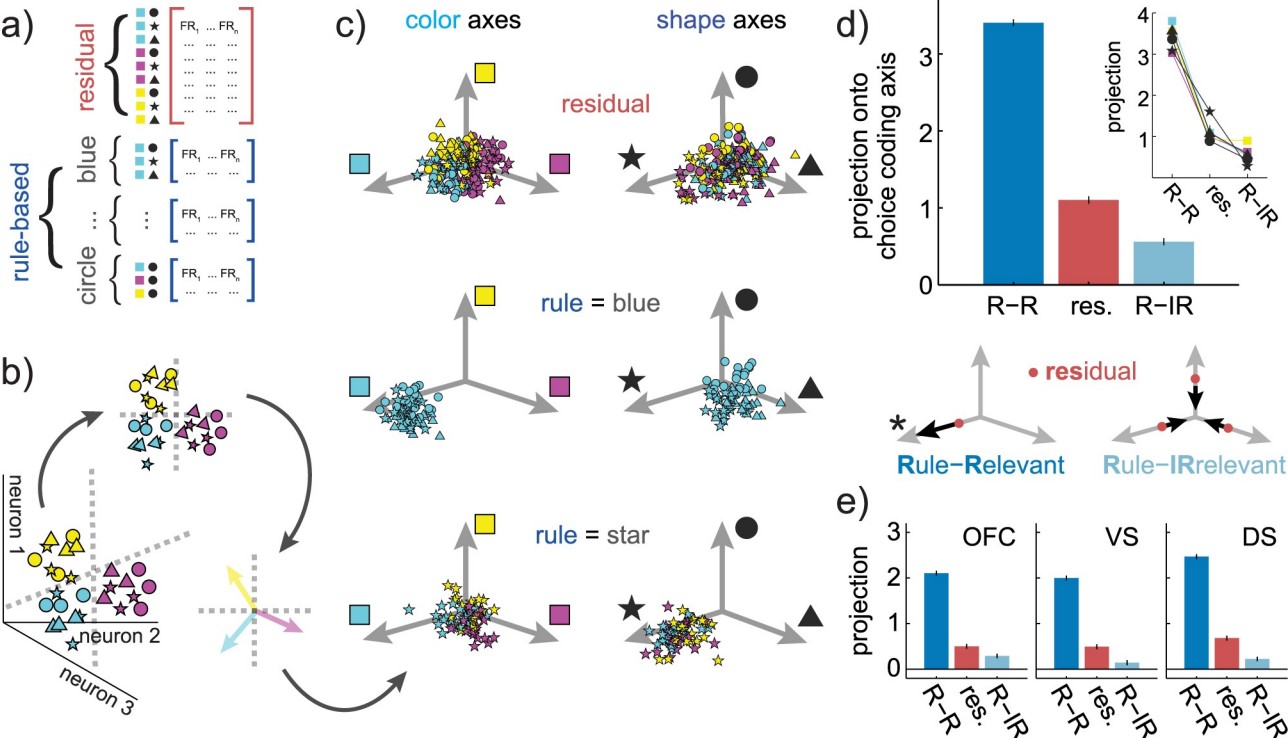

**Fig 7. Changes in choice representations within a choice-predictive subspace.** (A) To determine how adhering to a specific rule changed choice representations within a choice-predictive subspace, pseudopopulations are constructed for each rule-based and residual state. (B) Choice-predictive subspaces are identified with multiple logistic regression. This finds the separating hyperplanes (dotted gray lines) that best separate each class of choices in neuron-dimensional space. The cartoon illustrates 2 color hyperplanes, one separating magenta choices from other choices and one separating yellow choices from other choices. Pseudotrial activity is then projected into the subspaces defined by the color and shape hyperplanes. Within these subspaces, specific vectors correspond to the coding dimensions in neural activity that predict whether the choice will include each color or shape. These coding dimensions are included as reference vectors in the following panel. (C) Distribution of residual (top), color rule (middle), and circle rule (bottom) pseudotrials in each of the color (left column) and shape (right column) subspaces. (D) The projection onto the coding dimensions of the chosen features for residual decisions (red) and rule-based decisions (blue), with the latter separated according to whether the coding dimension is rule-relevant (dark blue, R-R) or rule-irrelevant (light blue, R-IR). Error bars = STE across pseudotrials. (Inset) Average projections for each color and shape rule, plotted separately. (Bottom) A cartoon illustrating the central effect: Compared to residual trials (red dots), rule-based decision-making pushes choice representations further along the rule-relevant axis (*), while compressing the rule-irrelevant subspace. (E) Same as D, plotted separately for OFC, VS, and DS. Data: https://doi.org/10.6084/m9.figshare.13139450.v1. DS, dorsal striatum; FR, firing rate; OFC, orbitofrontal cortex; STE, standard error; VS, ventral striatum.

56.2]). This implies that rule adherence did affect how choices were represented along choice-predictive coding dimensions but that changes in choice-predictive coding dimensions could not be fully explained by encoding of the rule identity.

To understand how choice representations changed along choice-predictive coding dimensions, we projected each pseudotrial into the subspace defined by these coding dimensions. Formally, this meant that we found a low-dimensional projection of neural activity where position corresponded to the decoded probability of choice. For illustration, the 3 choice color coding dimensions were grouped into color subspace, while 3 choice shape coding dimensions were grouped into shape subspace (**Fig 7C**). Projections onto chosen feature coding dimensions were substantially higher than projections onto unchosen feature coding dimensions (chosen mean = 1.69, range across choice categories = [1.55, 1.81]; unchosen mean = −1.54, range = [−1.64, −1.35]; $p < 0.0001$, t(1,3238) = 67.29, 2-sample $t$ test, 95% CI = [3.13, 3.32]; **Table D in S1 Text**). Thus, the projection of each pseudotrial onto these choice-predictive axes did indeed predict choice.

The magnitude of choice coding dimension projections depended on the subjects' goal: whether they chose the stimulus because the category was relevant to the rule, irrelevant to the rule, or the result of residual-state decision-making (**Fig 7D**; 2-way ANOVA, main effects of residual versus rule-based: $F(1, 1077) = 28.1$, $p < 0.0001$, main, nested effect of whether the feature was relevant to the rule: $F(1, 1077) = 229.4$, $p < 0.0001$). The representation of rule-relevant choice features was pushed furthest along the choice-predictive axis (mean = 3.40, range across categories = [3.03, 3.59]; sig. greater than rule-irrelevant: $p < 0.0001$, $t(1,718) = 44.1$, 95% CI = [2.72, 2.97], 2-sample $t$ test; sig. greater than residual: $p < 0.0001$, $t(1,718) = 35.1$, 95% CI = [2.17, 2.43]). The next largest projection was for the residual choices (mean = 1.10, range across categories = [0.89, 1.62]; sig. greater than rule-irrelevant: $p < 0.0001$, $t(1,718) = 8.13$, 95% CI = [0.41, 0.67], 2-sample $t$ test; all pairwise comparisons survive correction for multiple comparisons). Finally, rule-irrelevant features had the smallest coding dimension projection, meaning that they were the least discriminable (mean = 0.56, 95% CI = [0.29, 0.90]). These same effects were apparent in all 3 regions individually (**Fig 7E**, **Table E and F in S1 Text** **and S6 Fig**). Thus, relative to how choices were represented during residual-state decisions, rule-based decision-making expanded the rule-relevant dimensions of choice representational space, but compressed the rule-irrelevant dimensions across a distributed network of regions linked to decision-making.

## Changes in single-neuron tuning functions suggest a combination of early and late selection

Together, the results suggest that rule-based decision-making expands the representation of rule-relevant choice dimensions, while compressing dimensions that are rule-irrelevant. Superficially, this appears consistent with the influential idea that executive processes may enhance the processing of relevant stimulus dimensions via some kind of attentional gate or template [20]. However, rules warp signals related to the developing choice, not the stimulus, and changes in choice representations could be due solely to late effects in the decision-making process itself [22,35]. Attention, conversely, is the selection of 1 stimulus for enhanced perceptual processing at the expense of another [26,63,64]. Therefore, to determine rules modulated visual attention, we next asked whether rules altered neural tuning for stimuli.

Here, 3 options were presented sequentially in advance of the decision. These options were composed of random combinations of color and shape, which meant that we could measure the neural response to every stimulus color and shape on every trial, during the three 400-ms epochs when each option was presented alone on the screen. Asynchronous presentation is not typically ideal for measuring the effects of visual attention, because attentional enhancements are largest when multiple stimuli are presented in competition [65]. However, our hypothesis is that the different features of the same stimuli would be in competition for attentional selection. Based on what we know about feature-based attention from studies that used unidimensional stimuli [26,28], we reasoned that feature-based attention would enhance neuronal selectivity for selected, rule-relevant stimulus features at the expense of selectivity for unselected, rule-irrelevant features. Indeed, this is what we saw.

A small, insignificant number of individual cells were tuned for either the color or shape of the option stimuli (2-way ANOVA with main effects of color and shape fit to each neuron, 10.2% of neurons, 43/442, were significant for either color or shape at $p < 0.05$, not significantly greater than the expected false positive rate of 9.75%). However, the rule relevance of the feature dimension still modulated this minimal tuning (**Fig 8A**; generalized linear model, see Methods; significant interaction between feature preference and feature relevance: beta = 0.01, $p < 0.05$; significant main effect of feature tuning: beta = 0.03, $p < 0.0001$; nonsignificant

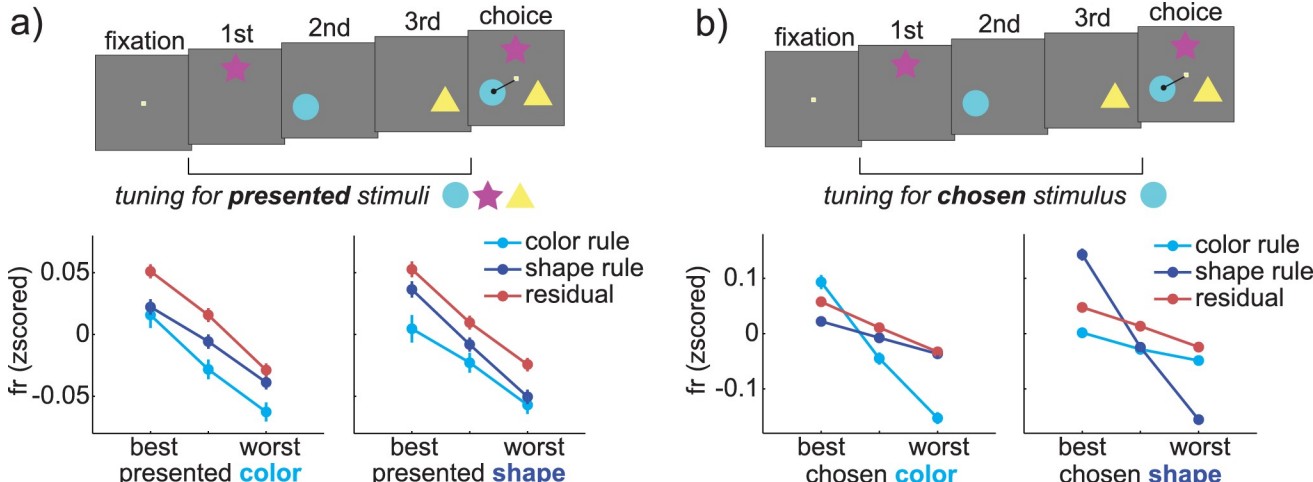

**Fig 8. Effects of rules on neuronal tuning for stimulus features and choice features.** (A) Neuronal tuning for the color (left) and shape (right) of presented stimuli during the three 400-ms epochs when these stimuli were on the screen, plotted separately for color rule, shape rule, and residual trials. "Best" and "worst" refer to the stimulus features associated with the highest or lowest firing rate, respectively, within neurons. (B) Same as A, but sorted by choice features, rather than stimulus features. Note the change in the y-axis needed to illustrate this substantially larger effect. Error bars = standard errors across 422 neurons; these may be smaller than the symbols. Data: https://doi.org/10.6084/m9.figshare.13139450.v1.

main effect of feature relevance: beta = −0.02, $p = 0.07$). Within each feature dimension, rule relevance increased the difference in firing rate between the response to the best feature (i.e., the stimulus shape or color associated with the highest firing rate across all epochs) and the response to the worst feature (lowest firing rate; shape rule: color tuning = 0.061 z-scored firing rate units ± 0.092 STD, shape tuning = 0.087 ± 0.107, mean pairwise difference = 0.026, paired $t$ test: $p < 0.0002$, t(1,421) = 3.92, 95% CI = [0.013, 0.039]; color rule: color tuning = 0.078 ± 0.150 STD, shape tuning = 0.062 ± 0.174, mean pairwise difference = 0.017, paired $t$ test: $p < 0.05$, t(1,421) = 2.42, 95% CI = [0.0004, 0.033]; both effects survive correction for multiple comparisons). No difference between color and shape tuning was observed during residual trials (color tuning = 0.080 ± 0.096, shape tuning = 0.077 ± 0.115; mean pairwise difference = 0.003, $p > 0.6$). Changes in stimulus tuning were quite modest in magnitude and were not observed within individual regions when treated in isolation. However, early changes in sensory encoding did exist and thus could have contributed to changes in choice representations.

Next, we directly compared the magnitude of rule-based changes in tuning for stimuli with the magnitude of rule-based changes in tuning for choice. This means we performed the same procedure as the last analysis, but now looked at how rules changed how neurons were tuned to features of the ultimate choice, instead of to the features of the stimuli, during this same set of 3 epochs per trial. As expected, rule relevance modulated neural tuning to features of the choice (**Fig 8B**; significant interaction between feature preference and feature relevance: beta = 0.21, $p < 0.0001$; significant main effect of feature tuning: beta = 0.027, $p < 0.0001$; significant main effect of feature relevance: beta = 0.11, $p < 0.0001$). Again, within each feature dimension, rule relevance increased the difference in firing rate between the response to the best feature (highest firing rate) and the response to the worst (lowest firing rate; shape rule: color tuning = 0.059 z-scored firing rate units ± 0.132 STD, shape tuning = 0.296 ± 0.320, mean pairwise difference = 0.24, paired $t$ test: $p < 0.0001$, t(1,421) = 14.66, 95% CI = [0.20, 0.27]; color rule: color tuning = 0.245 ± 0.324 STD, shape tuning = 0.047 ± 0.154, mean pairwise difference = 0.196, paired $t$ test: $p < 0.0001$, t(1,421) = 10.20, 95% CI = [0.16, 0.23]; both effects survive correction for multiple comparisons). Thus, the effects of rule relevance on

choice feature tuning were an order of magnitude larger than the effect of rule relevance on stimulus feature tuning, within in the same restricted epochs. Together, these results reinforce the central result that rule-based decision-making expands the representation of choice dimensions that are rule-relevant, while compressing dimensions that are rule-irrelevant. However, they also suggest that early changes in selectivity for stimulus features could contribute to these effects.

## Discussion

We found that rule-based decisions made more efficient use of limited neural resources than non-rule–based decisions. Fewer spikes were needed to generate rule-based choices across a distributed network of decision-making regions. Further, although firing rates were reduced during rule-based choices, information about the features of the chosen option was increased —suggesting rule-based choices were more metabolically efficient than non-rule–based choices. Rules were well learned before data collection began, so these results do not exclude the possibility that acquiring a rule has substantial metabolic demands, but they do resonate with the idea that we use rules, in part, because applying them reduces the energetic costs of decision-making. One way that implementing a rule could improve the efficiency of decision-making is by changing how decision-making problems are represented in the brain [15–17]. Indeed, we found distributed changes in neuronal representations that are broadly consistent with this idea. In 3 brain regions, OFC, VS, and DS, adhering to a rule warped the geometry of neural decision-making space in 2 ways: It expanded neuronal choice representations along rule-relevant choice coding dimensions, while compressing representations along rule-irrelevant ones.

In cognitive theory, simple policies like rules are thought to reduce the computational costs of coming to a decision largely because they eliminate the need to either represent or to perform computations on rule-irrelevant information [11,15,16,18,19]. However, in this study, the compression in rule-irrelevant coding dimensions was matched, if not exceeded, by expansion in rule-relevant dimensions. If there was some conserved decision-making quantity—like a fixed amount of evidence or value that must be accumulated to come to a choice—then the expansion of rule-relevant dimensions could be an inevitable consequence of compression in irrelevant dimensions. This is because eliminating the encoding of rule-irrelevant information would shift the choice process to overrepresent rule-relevant information in order to meet the fixed threshold for generating a decision. Of course, it is not clear that value-based decisions depend on an integrate-to-bound process [66,67], and bottlenecks at other stages of sensori-motor processing could also lead to this type of conservation of evidence. Alternatively, rule-relevant coding dimensions may be expanded because expansion facilitates critical rule-relevant computations, like the ability to classify the choice with respect to the current rule.

Representational changes during rule-based decision-making were remarkably similar across OFC, DS, and VS, suggesting that rules may have distributed effects, at least across the regions tested here. This means that we did not observe evidence for a handoff in control during rule-based decisions: There was no shift in the locus of choice-predictive information from VS and OFC (structures implicated in flexible, deliberative decision-making [28–32]) to DS (a structure implicated in automatized decision-making [25–27]). Although our results are not consistent with a functional handoff between these specific structures, we cannot rule out the possibility that there may be a functional handoff in other regions or the possibility that there was a shift in how the regions were communicating with each other. Further, we did observe several striking differences between regions, consistent with the results of previous studies that focused on the differences between these regions [47,49,51]. First, we found that population

activity in VS was more sparsely distributed across neurons during rule-based decision-making. Conversely, in OFC and DS, activity was sparser during residual decisions. Because the BOLD signal, in particular, may be sensitive to the total number of activated neurons [52], it is possible that this dissociation could explain why studies using BOLD imaging might see an increase in BOLD signal during rule-based decisions in OFC and DS, but a decrease in BOLD in VS [47,49,51]. Second, we found that the representational space of choices was more compact during rule-based decisions than residual decisions in OFC, while in VS, adhering to a rule expanded the space of choice representations—increasing the distance between both similar and dissimilar choices. Third, nearly 90% of neurons in OFC encoded the identity of the current rule, which is a striking proportion. This resonates with modern views that OFC is responsible for tracking the current state of dynamic environments [68] and causal results that suggest that OFC lesions disrupt learning that requires state inference [50]. However, note that approximately 80% of cells in VS and DS also signaled rule information, suggesting OFC is not the only region that encodes the current belief state of the animal.

The results we have collected are necessarily limited in scope to OFC, VS, and DS, but they do raise the question of how other regions would behave. This question is especially salient for 2 key dorsal executive regions, dorsal anterior cingulate cortex, and dorsolateral prefrontal cortex. These regions are often treated as the locus of the central executive, suggesting that they would be likely to serve a regulatory role over the regions in our study [20,69–72]. However, an alternative view is that these regions are part of a larger network specialized for converting sensory information to actions [73–77]. In that view, we would predict that these regions would show largely similar functions to OFC, VS, and DS. Indeed, a recent study by our lab suggests that dorsal anterior cingulate may also show rule-specific dimensional warping [78].

These results are broadly consistent with theoretical work on rule-based decision-making, but the residual class of decisions was biased. Many, but not all of the residual choices resembled a kind of exploratory hypothesis testing process, where the animals cycled through possible rules until the correct one was found. This behavior resembled a classic win–stay/lose–shift in the sense that animals avoided repeating features after losses but tended to repeat features after wins. However, because the stimuli were multidimensional and the most common response was to stay in 1 feature dimension while shifting in the other, a win–stay/lose–shift policy would have to be implemented at the level of hypotheses about what rule was correct, rather than at the level of stimulus responses to be successful in this task [79–81]. Because residual decisions were biased toward exploration, it seems reasonable to interpret these results as evidence that exploration decreases the efficiency of decision-making and warps choice representations to preserve information about choice identity. This could be important for learning if it facilitates later credit assignment through some as-yet-undiscovered mechanism. However, it is not clear that we would have expected these neural results from what we know about exploration. It is true that some previous studies have reported higher net activity during exploration compared to exploitation [82,83], but these studies conflated exploration with errors of task performance. Other studies, using different methods, do not report a net change in neural activity with exploration in decision-making regions, much less the kinds of protracted effects we show here [84–87]. An explore/exploit account also cannot explain why following a specific rule would enhance the representation of rule-relevant features at the expense of irrelevant ones. Nevertheless, future work is needed to whether these results are a feature of rule-based decision-making or, alternatively, are only apparent in contrast with exploratory decisions. It will also be critical to determine whether instructed/cued rules have similar effects on neural activity. Alternatively, the representational changes we observe here could be a special feature of rules acquired through experience.

The idea that cognitive processes warp how the brain represents information has support in both influential theories of prefrontal function [20] and modern views at the intersection of executive function and selective visual attention [59,64,88–90]. Our results strongly support the idea of representational warping: showing that rules not only gate which stimuli are represented in the neural code [91] but also how different stimuli are represented. This result is particularly significant because several previous single neuron studies did not report evidence that task demands can gate or warp the representation of low-level sensory features in the brain ([4,21–24], though compare with fMRI results [92,93]). Future work is necessary to determine whether the critical difference between our study and previous single unit studies is one of task design, analysis method, or the fact that this study targeted limbic regions commonly implicated in decision-making, rather than prefrontal or extrastriate regions linked to controlling or executing visual attention [21–24]. Certainly, one straightforward way to produce broadly distributed representational changes would be through a change in visual attention, either at the level of the extrastriate cortex [26–28] or in the prefrontal regions that direct featural attention [94]. Early feature selection could, in theory, alter the way that stimulus features are represented within every region that receives ascending visual information. We do report some evidence in favor of this idea here. However, like many attentional effects, early feature selection was modest, and some kind of nonlinear amplifying process would still be needed to transform modest biases in stimulus representations into substantial biases in choice representations. Further, similar effects have not been reported in regions more clearly implicated in visual attention [4,21–24], and it is still not entirely clear that feature-based attention can prioritize 1 feature of a stimulus at the expense of another [27–29,32–34].

A change in sensory encoding is not the only way to produce changes in how choice information is represented in neural populations [22,35]. The effects we report here could also be due to other selective mechanisms, including changes in the intrinsic dynamics of populations of decision-making neurons [22,35], an adjustment in neuronal gain coming from dorsal anterior cingulate [77,95], or amplification through recurrent interactions between these regions and/or through the thalamus [96]. One particularly compelling possibility is the idea that they reflect a category bias that OFC, VS, and DS may share with or inherit from dorsolateral prefrontal cortex [20]. Prefrontal neurons can flexibly categorize objects with respect to arbitrary rules, meaning that neural responses to sample stimuli are more discriminable according to rule-relevant category boundaries than irrelevant boundaries [97,98]. Here, we find broadly similar effects in OFC, VS, and DS in a very different type of task, suggesting that these representational changes (1) are more broadly distributed than was previously known; and (2) occur spontaneously during rule-based decision-making. Of course, categorization is not necessary for good decision-making, even in this task, and we are more than capable of representing every dimension of these decision-making problems [22], so it is interesting to consider the possibility that these signals reflect the outcome of some kind of categorization process. Regardless of mechanism, representational warping could be a powerful way to facilitate good decision-making: making it easier to classify options with respect to the attributes that matter and thereby reducing the complexity and costs of decision-making.

## Methods

### Ethical statement

All animal procedures were approved by the University Committee on Animal Resources at the University of Rochester (protocol #101112, UCAR-2010-169) and were designed and conducted in compliance with the Public Health Service's Guide for the Care and Use of Animals. Animals received appropriate analgesics and antibiotics after all procedures and chambers

were kept sterile with regular antibiotic washes and sealed with sterile caps. All data presented here were collected previously and have been used in earlier work [6–8,76]. All analyses presented here are new.

## Surgical techniques

Two male rhesus macaques (*Macaca mulatta*) served as subjects. Standard surgical techniques, described previously [99], were used to implant a small prosthesis for holding the head and Cilux recording chambers (Crist Instruments, Hagerstown, Maryland, United States of America). Chamber positions were verified by magnetic resonance imaging with the aid of a Brainsight system (Rogue Research, Montréal, Quebec, Canada). Neuroimaging was performed prior to surgery at the Rochester Center for Brain Imaging on a Siemens 3T MAGNETOM Trio Tim using 0.5-mm voxels.

## Electrophysiological techniques

OFC (115 neurons), VS (103 neurons), and DS (204 neurons) were approached through a standard recording grid (Crist Instruments) using a standard atlas to define OFC (Area 13M), VS (the core of the nucleus accumbens), and DS (both the caudate and medial part of the putamen). Specific coordinates of each recording site have been described previously [7]. OFC recordings were made at gray matter sites between 0 and 9 mm above the ventral surface, lateral to the medial orbital sulcus, and between 29 and 36 mm rostral of the coronal plane containing interaural zero. VS recordings were made at gray matter sites between 0 and 8 mm from the ventral surface of striatum, between 0 and 9 mm lateral to the midline plane, and between 28 and 21 mm rostral of the coronal plane containing interaural zero. DS recordings were made at gray matter sites more than 8 mm from the ventral surface of striatum, between 0 and 9 mm lateral to the midline plane, and between 28 and 21 mm rostral of the coronal plane containing interaural zero. Recording locations and the locations of significant anatomical boundaries (the ventral surface of striatum, the medial orbital sulcus) were verified through a combination of Brainsight (Magnetic Resonance) guidance and listening for characteristic sounds of white and gray matter while lowering the electrodes, which in all cases matched the boundaries predicted by the Brainsight system.

During each session, between 1 and 4 single electrodes (Frederick Haer, Bowdoin, Maine, USA; impedance range 0.8 to 4 M) were lowered using a microdrive (NAN Instruments, Nazareth Illit, Israel) until the waveforms of single neuron(s) were isolated. Neurons were selected for study solely on the basis of the quality of isolation, not on task-related response properties. Cells were isolated and recorded with a Plexon system.

## General behavioral techniques

Animals were habituated to laboratory conditions and then trained to perform oculomotor tasks for liquid reward before training on the task. Previous training history for these subjects included 2 types of foraging tasks, intertemporal choice tasks, 2 types of gambling tasks, and a basic form of a reward-based decision task [100–103]. Eye position was sampled at 1,000 Hz by an infrared eye-monitoring camera system (SR Research, Kanata, Ontario, Canada). Stimuli were controlled by a computer running MATLAB (The MathWorks, Natick, Massachusetts, USA) with Psychtoolbox and Eyelink Toolbox. Visual stimuli were presented on a computer monitor placed 57 cm from the animal and centered on its eyes. A standard solenoid valve controlled the duration of juice delivery. The relationship between solenoid open time and juice volume was established and confirmed before, during, and after recording.

## Behavioral task

This task has been described in detail previously [6,7,39]. This present task is a version of the CSST, an analog of the WCST that was developed for use in nonhuman primates [40]. Task stimuli are similar to those used in the human WCST, with 2 dimensions (color and shape) and 6 specific rules (3 shapes: circle, star, and triangle; 3 colors: cyan, magenta, and yellow; **Fig 1A**). Choosing a stimulus that matches the current correct feature (e.g., any blue shape when the correct feature is blue; any color of star when the correct feature is star) results in visual feedback indicating that the choice is correct (a green outline around the chosen stimulus) and, after a 500-ms delay, a juice reward. Choosing a stimulus that does not match the correct feature results in visual feedback indicating that the choice is incorrect (a red outline around the chosen stimulus), and no reward for the trial.

The correct feature was fixed for block of trials. At the start of each block, the correct feature was drawn randomly. Blocks lasted until monkeys achieved 10, 15, 20, or 30 correct responses that matched the current rule. The number was fixed within session and depended on the subject's behavior; in practice, it was almost always 15. Because the block length was a fixed number of correct trials, that blocks lasted for a variable number of total trials, determined by both how long it took monkeys to discover the correct feature and how effectively monkeys exploited the correct rule, once discovered. Block changes were uncued, although reward omission for a previously rewarded option provided unambiguous information that the reward contingencies had changed.

On each trial, 3 stimuli were presented asynchronously. One stimulus was presented at each of 3 locations on the screen. The color, shape, position, and order of stimuli were randomized. Stimuli were presented for 400 ms and were followed by a 600-ms blank period. (The blank period is omitted in **Fig 1A** because of space constraints). Monkeys were free to look at the stimuli as they appeared, which they typically did [6]. After the third stimulus presentation and blank period, all 3 stimuli reappeared simultaneously with an equidistant central fixation spot. When they were ready to make a decision, monkeys fixated on the central spot for 100 ms and then indicate their choice by shifting gaze to 1 stimulus and maintaining fixation on it for 250 ms. If the monkeys broke fixation within 250 ms, they could either again fixate the same option or change their mind and choose a different option, although they seldom did so. Thus, the task allowed the monkeys ample time to deliberate over their options, come to a choice, and even change their mind, without penalty of error.

## General data analysis techniques

Data were analyzed with MATLAB. Two sample $t$ tests were used for all pairwise statistical comparisons except for the mutual information analyses because these data visibly deviated from the normality assumptions of the test. Neural activity was analyzed in the fixed 3,350-ms epoch that started at the presentation of the first option and ended just before feedback was provided. Firing rates were z-scored within neurons for all analyses, except in direct comparisons of total firing rate across rule-based and residual decisions (e.g., **Fig 2**), where firing rates were centered, but not scaled. All pseudopopulation results are reported for a single, randomly seeded pseudopopulation, but were confirmed with (1) different random seeds; and (2) bootstrap tests across 1,000 random seeds. To correct for multiple comparisons when calculating the proportion of neurons tuned for different task variables, we used a 1-sided binomial test against the expected false positive rate. Elsewhere, we used the Benjamini–Hochberg procedure [104]. For independent tests, this is guaranteed to produce a type I error rate at or below the family-wise rate, set at 0.05 by convention.

## Identifying information-maximizing choices

We determined how much information would be gained from different choice strategies with a model that used the recent history of rewards and choices to estimate the likelihood that each feature was the correct one [39]. The influence of past outcomes and choices decays exponentially fast [105], so we took advantage of the fact that the last trial has the single largest influence on choice to construct a simplified model with a 1-trial history. Assuming all possible histories of choices and rewards before the last trial ($X_{1:t-2}$), we initialize a uniform prior that each feature ($f$) of the $N_f$ features is the correct feature ($f^*$):

$$\pi(f = f^* | X_{1:t-2}) = \frac{1}{N_f}$$

After a choice is made at time $t-1$, we calculate the likelihood that the chosen feature was correct in a reward-dependent fashion. If the choice is rewarded (r = 1), the likelihood is:

$$\pi(f = f^* | r_{-1} = 1) = \begin{cases} \frac{1}{2}, if\ choice = f \\ 0, otherwise \end{cases}$$

If the choice is not rewarded, the likelihood is:

$$\pi(f = f^* | r_{-1} = 0) = \begin{cases} 0, if\ choice = f \\ \frac{1}{N_f - 2}, otherwise \end{cases}$$

Multiplying the prior and likelihood and renormalizing into a valid probability distribution gives the posterior probability that each feature is best after $t-1$, which is also the new prior at the start of the residual-state trial $t$. Because we knew, on average, whether or not the monkeys were rewarded for the last choice before a residual decision, we calculate the prior for residual decisions by taking an average of these 2 likelihoods, weighted by the probability that they occurred:

$$p(f = f^* | r_{-1}) = p(r_{-1} = 1)p(f = f^* | r_{-1} = 1) + p(r_{-1} = 0)p(f = f^* | r_{-1} = 0)$$

$$\pi(f = f^* | X_{1:t-1}) = \pi(f = f^* | X_{1:t-2})p(f = f^* | r_{-1})$$

To determine the information that could be gained from various choices, we first calculate the entropy of the prior:

$$H_t = -\sum \pi(f = f^* | X_{1:t-1})log_2\pi(f = f^* | X_{1:t-1})$$

Then, we calculate the posterior distribution that we would observe after each possible choice. We first calculate the likelihoods, as described above, for all possible combinations of choice ($c_t$) and reward outcome ($r_t$), then multiply each likelihood with the prior to get all possible posterior distributions after the choice at time t:

$$\pi(f = f^* | X_{1:t-1}, r_t, c_t) = \pi(f = f^* | X_{1:t-1})p(f = f^* | r_t, c_t)$$

There were 2 possible outcomes of each choice—one where the animal would be rewarded, and one where they would not—and the likelihood of these futures differed systematically across the different choices. Therefore, the posterior entropy for each choice is an average of

these possible futures, weighted by their likelihood of occurring:

$$\hat{H}_{t+1|c_t} = -\sum_{r\in[0,1]} p(r_t|c_t)\pi(f=f^*|X_{1:t-1}, r_t, c_t)log_2\pi(f=f^*|X_{1:t-1}, r_t, c_t),$$

where the probability of reward for each choice is just the probability that the choice will include the best feature:

$$p(r_t|c_t) = \sum_{f\in c_t} p(f=f^*)$$

The information maximizing choice is then one that would cause the largest drop in the model's uncertainty about what feature is correct. That is, it would be the choice, c, that maximizes:

$$\text{info gain} = H_t - \hat{H}_{t+1|c_t}$$

The reward-maximizing choice, conversely, is the one that maximizes the probability of reward. We calculated these values via simulating sequences of choices, then calculating belief states, belief state entropy, and estimated reward probabilities. In order to ensure that information would be computable, small amounts of noise, $|N(0, 10^{-4})|$, were added to all 0s in the belief states before these were renormalized into valid probability distributions. Choices were then classified as differing in 0, 1, or 2 stimulus features from the last choice, and the average information gain across these reward history classes, weighted by their relative frequency during residual choices, is illustrated in the inset of **Fig 1G**. That is:

$$\text{info gain} = p(r_{t-1}=0) \cdot \text{info gain}_{|r_{t-1}=0} + p(r_{t-1}=1) \cdot \text{info gain}_{|r_{t-1}=1},$$

where the $p(r_{t-1}=1)$ is simply the observed probability that the monkeys were rewarded on the trial immediately before a residual-state choice (i.e., 0.52).

### Hidden Markov model (HMM)

An HMM was used to infer the latent goal state underlying each choice. This model allowed us ask whether, for example, each blue-star choice was made because (1) it was a star; (2) it was blue; or (3) the monkey was searching for the correct feature or performing some other non-rule–based computation. We have previously used this method to identify latent goal states in other tasks [85] and described how this model was developed for this task [39].

In the HMM framework, choices ($x$) are "emissions" that are generated by an unobserved decision process that is in a latent, hidden state ($z$). Each hidden state has some observation model, which dictates the probability of choosing each option when the process that state. For example, the blue-rule state had an observation model where the probability of choosing the blue option was 1, but the probability of choosing a non-blue option was 0:

$$p(x_t = n|z_t = rule_i) = \begin{cases} 1, \text{if } n \text{ matches } rule_i \\ 0, \text{otherwise} \end{cases}$$

Conversely, because we do not know what monkeys would choose in a residual state, we default to the maximum entropy distribution. Thus, the observation model for any of the $N = 3$ choice options ($n$) during residual states is:

$$p(x_t = n|z_t = residual) = \frac{1}{N}$$

Inverting the emissions model allows us to make inferences about the state the animal was in at the time when they made the choice. For example, a blue-star choice can only be due to a blue rule, a star rule, or a residual choice, but not to a magenta rule. We can refine this inference further by taking the temporal structure of the hidden states into account.

Because the model is Markovian, the probability that monkeys will be in any state z at time t depends only on the immediately preceding state (**Fig 1C**), that is, the model assumes that states are conditionally independent of all previous states and choices, given observation of the most recent state:

$$p(z_t|z_{t-1}) = p(z_t|z_{1:t-1}, x_{1:t-1})$$

The temporal dependencies between states are represented by a stochastic matrix known as a "transition matrix," whose entries are the 1-time-step probability of transitioning from each state to every other state, including itself. We reasoned that monkeys could not divine the new rule following a change point and instead had to explore to discover it, so direct transitions between different rule states were fixed at 0 in the model. For the same reason, the monkeys were also assumed to start in the residual state on the first trial each day. The structure of the transition matrix is illustrated in Fig 1C.

Transition parameters were tied across the 6 rule states, meaning that the model ultimately had only 2 free parameters: the probability of staying in the residual state and the probability of staying in any of the 6 rule-based states. Fitting these 2 parameters allows the model to account for the natural temporal dynamics of the behavior. For example, rule-based states had a strong probability of transitioning to themselves, so they tended to be occupied for many trials in a row, while the residual states tended to be briefer. We can use these insights to refine our inference about the hidden state for our blue-rule choice. If a blue star is chosen now, and the last choice was a blue circle (due either do a blue rule, a circle rule, or a residual state), then the most probable explanation, given these temporal dynamics, is that we are currently in a blue rule state. The next most probable explanations are that we just observed 2 residual-state choices in a row, or a transition from a star-rule to a residual state. However, if the next choice is a blue option or 2 trials back was also a blue option, then these alternative explanations become less and less likely.

These 2 parameters were fit to each individual session via expectation–maximization using the Baum Welch algorithm [106,107]. This algorithm finds a (possibly local) maxima of the complete data likelihood, which is related to the joint probability of the hidden state sequence and the sequence of observed choices. The algorithm was reinitialized with random seeds 100 times, and the model that maximized the observed (incomplete) data log likelihood was ultimately taken as the best for each session. We then used the Viterbi algorithm to identify the most probable a posteriori latent state underlying each choice [107], given the entire sequence of choices and the parameters of the fitted model.

## Model comparison

We used model comparison to determine whether the state labels inferred from the HMM better explained neural activity than other approaches we could have used to separate the different classes of trials. Firing rates of each neuron i were modeled as Poisson distributed, with the mean rate dependent on whether or not each trial t was rule-based:

$$\mathrm{fr}_{i,t} = e^{\beta_0 + \beta_1(\mathrm{rule}_t)}$$

The logical "rule" was defined in one of 3 ways: (1) rule versus not-rule (i.e., residual-state) as inferred from the HMM; (2) correct versus incorrect choices; and (3) correct responses 5 or

more trials after the block change versus other choices. We then calculated the log likelihood of the data under each of the 3 models and, from that, their Akaike information criterion (AIC) and Bayesian information criterion (BIC) values (Matlab: aicbic). AIC and BIC provide measures of the relative likelihood of sets of models, but their units are arbitrary, so they are transformed into relative likelihoods (AIC or BIC weights) for quantitative comparison.

## Sparsity and the Gini index

The Gini index is a measure of the sparsity of the distribution of some resource (in this case, spikes) across a population (in this case, neurons). Here, a sparse distribution is one in which a disproportionate number the total population spikes were emitted by a small number of neurons. The Gini index has a few nice properties for measuring spike sparsity in that it is invariant to both the number of spikes and the number of neurons and it is sensitive to the addition of long-tail outliers, both in terms of very low and very high firing neurons (53).

Given a sorted vector of mean firing rates for n neurons (a sorted pseudotrial, see below), x = [$x_1$, $x_2$ . . . $x_n$], where $x_1 \leq x_2 \leq x_n$, the Gini index is calculated as:

$$Gini\left(\overrightarrow{x}\right) = 1 - 2\sum_{k=1}^{N} \frac{x_k}{\|x\|_1}\left(\frac{N-k+\frac{1}{2}}{N}\right)$$

To give an intuition for this measure, the Gini index quantifies the area under the curve in a plot of the cumulative total population spikes against the cumulative total number of neurons (**Fig 3A**). If each neuron contributed equally to the total spike count, this plot would follow unity and the Gini index would equal 0.5. As more and more neurons are silent (equivalently, as more of the spikes are concentrated in a smaller number of neurons), this curve will deviate more from the unity line and the Gini index will approach 1. The Gini index was calculated for each pseudotrial, using the same pseudotrial matrix illustrated in **Fig 5**.

## Mutual information

To estimate the amount of choice information in single neurons, we calculated mutual information between firing rate (r) and choice features (c) as:

$$I(R; C) = \sum p(c)p(r|c) * log_2\left(\frac{p(r|c)}{p(r)}\right)$$

Firing rates were discretized via quantile binning within neurons to allow us to calculate these probabilities directly. Two firing rate bins per neuron were used for 2 reasons (though the results were similar with more firing rate bins). First, it allowed us to include all the cells. There were a small number of very sparsely firing neurons whose spike counts could not be quantile binned into more than 2 bins (i.e., they did not fire during this epoch approximately half the time, but, when they did, fired between 1 and 3 spikes). Second, it maximized the accuracy of our probability calculations. To perform these analyses, we had to estimate a minimum of 36 probabilities (i.e., combinations of 2 choice types, 9 choice identities, and at least 2 firing rate bins) from a finite sample of trials. As the number of bins increases, the number of observations per bin goes down, and it becomes increasingly difficult to determine whether a bin is empty because its probability is truly 0 or because we simply did not observe it in an increasingly small sample.

There were fewer residual trials than rule-based ones, and mutual information estimates are systematically inflated when the number of observations is low [54]. We controlled for this limited-sampling bias in 2 ways. First, we randomly downsampled the rule-based trials to

match the count of residual trials and report the results of this downsampled distribution in the text and **Fig 4**. Second, we calculated the expected mutual information under the hypothesis of no relationship between choice and firing rate in each case, including these as reference points in the plot. This was done by randomly shuffling labels with respect to the firing rate vectors.

## Pseudopopulation choice representations

To estimate how choice representations at the population level changed with rules, we combined all the neurons into pseudopopulations. That is, we treated separately recorded neurons as if they were simultaneously recorded [22,60–62]. Though this pseudopopulation approach does not allow us to reconstruct the covariance structure between simultaneously recorded neurons, it can still be useful for generating first order insights into how population activity changes across various conditions.

Within each task condition (combination of chosen color, shape, and state), firing rates from separately recorded neurons were randomly drawn with replacement to create a pseudotrial firing rate vector for that task condition, with each entry corresponding to the activity of 1 neuron in that condition. Together, these pseudotrial vectors were stacked into the trials-by-neurons pseudopopulation matrix. Twenty pseudotrials were drawn for each condition, based on the observation that approximately 75% of conditions had at least this number of observations (mean per neuron per condition = 28, median = 26). If a small number of conditions were missing for a particular neuron (<5), we imputed the neurons' mean firing rate. If a larger number of conditions were missing, the neuron was excluded from population analyses. Neurons were also excluded if their mean firing rate was less than 2 spikes/s [22]. Four of 422 neurons were excluded by these criteria, all of which were in VS. Pseudopopulation results reported in the main text come from a single pseudopopulation and were confirmed by bootstrap tests across 1,000 randomly re-seeded pseudopopulations (**Table A–F in S1 Text**).

We constructed a total of 3 different pseudotrial matrices, as illustrated in **Figs 5, 6 and 7**. These differed only in how we defined "state." The first pseudopopulation (**Fig 5**) was constructed without respect to the specific rule that monkeys were using (i.e., all rule-based blue-star choices were combined, regardless of whether the rule was choose-blue or choose-star). We took this approach because we had no knowledge of which, if any, stimulus features were relevant during residual decisions. By creating a rule-based pseudopopulation without respect to the specific rule, we equated our knowledge about the 2 conditions and isolated any changes in choice representations during rule-based decision-making. The second pseudopopulation (**Fig 6**) included information about whether the rule was color-based or shape-based, but not any information about whether it was a circle, star, or triangle rule, for example. This allowed us to isolate the effects of rule domain on stimulus feature representations. Finally, the third pseudopopulation (**Fig 7**), included information about the specific rule that the monkeys were choosing (blue-rule or circle-rule). This allowed us to determine how specific rules affected rule-relevant and rule-irrelevant choice dimensions.

## Representational similarity analysis

To measure the representational similarity between different choices in a pseudopopulation, we first calculated the geometric mean choice representation for each combinations of choice identity and choice type (i.e., rule-based versus residual choices or color-rule versus shape-rule choices). Then, we measured the Euclidian distance between every pair of mean choice representations. This created a matrix of pairwise distances between choice representations (i.e., **Figs 5 and 6**) in arbitrary units, which was z-scored for all reported analyses. All analyses and z-

scoring excluded the distance between the chosen stimulus identity and itself, which was 0 by definition. Representational similarity matrixes are illustrated for 1 example pseudopopulation.

To determine whether there was significant structure in the representational similarity matrices shown in **Figs 5** and **6**, we fit ANOVAs to the upper triangular of the matrices (excluding the diagonal, which was all ones by definition). The ANOVA for **Fig 5** (rule-based versus residual choices) included 3 terms: (1) within versus between choice types; (2) within rule versus within residuals (nested in the within versus between predictor); and (3) sharing a feature versus not sharing a feature. The ANOVA for **Fig 6** (color rule versus shape rule) included a similar set of 3 predictors: (1) within versus between choice types; (2) within color rule versus within shape rule (nested in the within versus between predictor); and (3) sharing a color versus sharing a shape. In this second model, we omitted cells that matched both shape and color but not rule type because the predictor matrix was not full rank after removing the diagonal of ones. All predictors were modeled as categorical variables, and the models included all possible interaction terms.

## Category and identity tuning in single neurons

To determine whether choice category (versus identity) tuning was increased in single neurons during rule-based decision-making, we first quantified the extent of stimulus identity tuning in single neurons with an ANOVA. Here, we modeled the firing rate $Y$ on a given trial $k$ where choice $i$ was made as:

$$Y_{i,k} = \mu + \text{stimulus}_i + \epsilon_{i,k},$$

where stimulus$_i$ was identity-coded, meaning that each of the 9 unique possible combinations of color and shape were modeled independently. For each neuron, we then calculated the extent to which this model captured the variance in the neural data by calculating the $F$ test statistic of this ANOVA:

$$F_{cat} = \frac{\text{variance between identities}}{\text{variance within identities}}$$

Next, we did the same with a model that independently modeled the contribution of the chosen stimulus color and shape to firing rate:

$$Y_{i,k} = \mu + \text{color}_i + \text{shape}_i + \epsilon_{i,k}$$

In this model, color and shape effects are modeled as independent and additive. That is, this model assumes that there is no encoding of stimulus identity beyond what can be explained by separately knowing how a neuron responds to chosen color and shape. Again, we calculated how well this model captured firing rates via its $F$ test statistic:

$$F_{cat} = \frac{\text{variance between categories}}{\text{variance within categories}}$$

These 2 models were fit separately to each neuron during rule-based and residual decisions, respectively. We then calculated the relative amount of stimulus identity and stimulus category tuning during the 2 types of decisions with a "identity–category index," which is a ratio of the $F$ test statistics of the choice identity and choice category models:

$$\text{index-category} = \frac{F_{id}}{F_{cat}}$$

Here, a value greater than 1 indicates that independently modeling each choice's identity improved model fit, compared to modeling only choice categories. Conversely, a value equal to or less than 1 indicates that choice category information was sufficient to explain the variance in firing rate during decision-making.

## Choice predictive subspace analyses

To identify choice-predictive subspaces, we used a form of targeted dimensionality reduction based on multinomial logistic regression [85]. Targeted dimensionality reduction is a class of methods for re-representing high-dimensional neural activity in a small number of dimensions that correspond to variables of interest in the data [22,108–110]. Thus, unlike principle component analysis—which reduces the dimensionality of neural activity by projecting it onto the axes that capture the most variability in the data—this approach reduces dimensionality projecting activity onto axes in neuronal state space that encode task information or predict behavior.

Here, we were interested in how rule-based decision-making changed how choices were represented, so we first identified the axes in neural activity that predicted choice. The design of this task allowed us to dissociate choice-predictive axes from axes that encoded rule information because the same choice could be generated in 3 ways: via color-rule, shape-rule, or non-rule–based computations. Because these occurred in roughly equal proportion (33% of trials were residual choices, while 67% were distributed evenly across shape and color rules), we could identify the axes that predicted choice, regardless of why this choice was made.

We used multinomial logistic regression to find the separating hyperplanes in neuron-dimensional space that best separated choices to 1 category (i.e., blue) from other choices (i.e., not blue). Formally, we fit a system of 6 logistic classifiers:

$$\log\left(\frac{p(choice = i|X)}{1 - p(choice = i|X)}\right) = X\beta_i,$$

where $X$ is the trials by neurons pseudopopulation matrix of firing rates, and $\beta_i$ is the vector of coefficients that best differentiated neural activity on trials in which a choice matching category $i$ is chosen from activity on other trials. The separating hyperplane for each choice $i$ is the vector ($a$) that satisfies:

$$a^T\beta_i = 0$$

Meaning that $\beta_i$ describes a vector orthogonal to the separating hyperplane in neuron-dimensional space, along which position is proportional to the log odds of choice. By projecting a pseudotrial vector x onto a coefficient vector $\beta_i$, we are re-representing that trial in terms of its distance from the separating hyperplane corresponding to choice category $i$. Projecting that trial onto all 6 classifiers, then re-represents that high-dimensional pseudotrial in 6 dimensions—each one corresponding to the likelihood that the choice will include that feature as predicted by the population activity. Because only decisions within a feature domain were mutually exclusive, the logistic classifiers naturally grouped into 2 sets: the color-category classifiers and the shape-category classifiers. These defined 2 subspaces in the neural activity: one in which trial position predicted choice color and one where it predicted choice shape (**Fig 7**).

Separating hyperplanes were fit via regularized maximum likelihood (ridge regression). Regularization helps reduce overfitting by penalizing models with large coefficients. The extent of this penalty—the regularization parameter λ—was set to the minimum value that maximized cross-validated classification accuracy (20-fold cross-validation, training on pseudotrial matrices constructed from 90% of the trials and tested on matrices constructed from the 10%

of trials that were held out; cross-validated accuracy evaluated at 25 log-spaced values for λ, range: $[0, 10^4]$). Nearly identical results were observed across a wide range of λ values, including 0. Rule-encoding vectors were identified through the same procedure as choice-encoding vectors, but the classifiers were trained to predict the color or shape of the rule, rather than the color or shape of the choice.

To determine whether rules had any effect on where neural activity lay in the choice predictive subspace, we fit an ANOVA with 2 terms: whether the decision was rule-based or residual decision, and then, nested within that, whether the axis was relevant to the rule, irrelevant to the rule, or if it was residual and we had no knowledge either way.

### Rule-effects on single-neuron tuning for choices and stimuli

In order to determine how rules affected neuronal tuning for stimuli and choices (i.e., **Fig 8**), we focused on the three 400-ms stimulus presentation epochs that occurred during each trial. Firing rate was z-scored (centered and scaled to unit variance within cells), then average firing rate was calculated within each condition, where conditions referred to all combinations of stimulus color and color or shape rule and stimulus shape and color or shape rule. We then fit the following generalized linear model to the set of neural responses:

$$\text{fr} = \beta_0 + \beta_1 S + \beta_2 R + \beta_3 (S \cdot R)$$

In this model, "S" coded whether the stimulus was the best (3), intermediate (2), or worst stimulus (1) for that cell, within that feature domain. (This was calculated within each cell, across all trials, without regard to state.) "R" reflected the relevance of the stimulus and was coded as 1 where the feature domain was relevant to the rule, and 0 otherwise. $\beta_1$ thus described the slope of the tuning for color and shape and is expected to be positive because the "best" stimulus was the one with the highest average firing rate by definition. $\beta_2$ described any offset in firing rate between rule-relevant and rule-irrelevant trials. Finally, $\beta_3$ was the critical term that described the sharpening or flattening of the tuning curves when that feature dimension was relevant to the animals' current state. To examine neural tuning for choices in the same epoch, we repeated this procedure, but now sorted trials according to the color or shape that would be chosen on that trial, rather than the stimuli on the screen.

### Supporting information

**S1 Fig. The HMM explains neural activity better than other approaches.** Related to Figs 1 and 2. Multiple approaches have been used to identify the rule-based decisions in tasks where the correct rule changes over blocks. The most common approach is to identify rule-based choices as the correct choices that occur within a block. Alternatively, often in tasks where rule changes are uncued, rule-based choices may be identified as the correct choices that occur after some initial burn-in or learning period (e.g., after 5 trials). Here, we instead modeled rules as the latent states underlying decisions in an HMM, then used the HMM to infer which decisions were most likely to be rule-based. To determine whether this approach was appropriate, we used model comparison to ask whether the rule labels inferred from the HMM (dark gray) better explained variance in neural activity than rule labels derived from other approaches (all correct choices within a block = light gray, all correct choices after a 5 trial burn-in period = middle gray). In model comparison, the model with the lowest AIC and BIC values is the preferred model. Within each region, AIC and BIC values were both lowest for the HMM, indicating that this approach explained the most variance in neural activity. The AIC and BIC weights for all the alternative approaches were less than $10^{-32}$, indicating very strong evidence that the HMM approach was the best. Data: https://doi.org/10.6084/m9.

figshare.13139450.v1. AIC, Akaike information criterion; BIC, Bayesian information criterion; DS, dorsal striatum; HMM, hidden Markov model; OFC, orbitofrontal cortex; VS, ventral striatum.
(TIF)

**S2 Fig. Residual-state choice patterns balance reward and information maximization.** (A) The expected information gain (left) and expected reward probability (right) for choosing an option that differs in 0, 1, or 2 dimensions from the previous choice, given that the last trial was rewarded (cyan) or not rewarded (gray). The blue dotted lines shown here reflect the combination of these 2 functions, given the typical reward history on residual-state trials (i.e., a weighted average of the 2 functions, where the weight reflects the fact that 52% of residual choices follow reward delivery). (B) The choice patterns illustrated in Fig 1G, now plotted separately for residual-state choices following reward omission (left) and reward delivery (right). Conventions are the same as Fig 1G. Arrows reflect the choices that would maximize information (black arrows) or rewards (blue arrows), given that reward history, as illustrated in the previous panel. (C) Same as B, for all choices after the first omitted reward after a block change. Data: https://doi.org/10.6084/m9.figshare.13139450.v1
(TIF)

**S3 Fig. Changes in choice representations across rule-based and residual-state decision-making across pseudopopulations.** Related to Fig 5C. Distribution of means across 1,000 bootstrapped pseudopopulations for the contrasts across the representational similarity matrix. Plotted separately for OFC (top row), VS (middle), and DS (bottom). Left column) If there is a change in how choices are represented between rule-based and residual decisions, then the mean distance between rule-based and residual choice representations (light gray, average of off-diagonal blocks) should be greater than the distance within decision types (dark gray, average of on-diagonal blocks). (Middle column) If there is a change in the total representational space between decision types, then the mean distance within rule-based decisions (blue, average of top left block) should differ from residual decisions (red, average of bottom right block). (Right column) If the categorical structure of the choices is represented, then the mean distance between stimuli that share features ("match," dark color) should be smaller than the distance between choices that do not share features ("no match," light color). This is calculated separately for both rule-based (blue) and residual (red) decisions. Dots = mean across pseudopopulations, error bars = 95% CI, asterisks = significant contrasts, all $p < 0.0001$. Data: https://doi.org/10.6084/m9.figshare.13139450.v1. DS, dorsal striatum; OFC, orbitofrontal cortex; VS, ventral striatum.
(TIF)

**S4 Fig. Cross-validated changes in choice representations across rule-based and residual-state decision-making.** Related to Fig 5B and 5C. (A) Distances between choice representations in nonoverlapping subsets of trials. Within each cell, trials were partitioned at random into 2 equal subsets, with the constraint that each trial type was represented at least once in each subset. Then, the 2 subsets were used to construct 2 independent pseudopopulations. We then performed the same analysis as Fig 5B but calculated distances between choice representations as the distance between the representation in 1 set and the representation in the other set. After in Fig 5B, this panel illustrates 1 example partition/pseudopopulation pair. (B) The same analysis as **S3 Fig**, but distances are calculated from nonoverlapping subsets of the data, as in panel A. Data is combined across regions. Dots = mean across pseudopopulations, error bars = 95% CI, asterisks = significant contrasts, all $p < 0.0001$. Specific contrasts, from left to right. Choice representations were closer to themselves within choice type than between choice

types (mean difference = 0.72, 95% CI = [0.40, 1.02], $p < 0.0001$, 1-sided bootstrap test). Mean representational difference was smaller during rule-based than residual choices (mean difference = 1.52, 95% CI = [1.11, 1.92], $p < 0.0001$). Residual choices that shared a feature were not closer together than choices that did not (mean difference = 0.01, 95% CI = [−0.31, 0.33], $p = 0.48$). However, rule-based choices that shared a feature were more similar than rule-based choices that did not share a feature (mean difference = 0.97, 95% CI = [0.68, 1.27]). Data: https://doi.org/10.6084/m9.figshare.13139450.v1
(TIF)

**S5 Fig. Changes in choice representations during color and shape rules across pseudopopulations.** Related to Fig 6C. Distribution of means across 1,000 bootstrapped pseudopopulations for contrasts across the representational similarity matrix in **Fig 6B**. Plotted separately for OFC, VS, and DS (top to bottom). (Left column) A change in representation between color-rule and shape-rule decisions would increase distance between representations measured across decision types (light gray), compared to within decision types (dark gray). (Right column) Mean distance between stimuli that share the rule-relevant feature (saturated color) and rule-irrelevant feature (dim color), during shape-rule (left column) and color-rule (right column) decisions. Dots = mean, error bars = 95% CI across 1,000 bootstrapped pseudopopulations, asterisks = significant contrasts, all $p < 0.001$ (see **Table C in S1 Text**). Data: https://doi.org/10.6084/m9.figshare.13139450.v1. DS, dorsal striatum; OFC, orbitofrontal cortex; VS, ventral striatum.
(TIF)

**S6 Fig. Changes in choice representations within a choice-predictive subspace, distribution of means across pseudopopulations.** Related to Fig 7E. The projection onto the coding dimensions of the chosen features for residual decisions (red) and rule-based decisions (blue), with the latter separated according to whether the coding dimension is rule-relevant (dark blue, R-R) or rule-irrelevant (light blue, R-IR). Plotted separately for OFC, VS, and DS. Dots = mean, error bars = 95% CI across 1,000 bootstrapped pseudopopulations. Data: https://doi.org/10.6084/m9.figshare.13139450.v1. DS, dorsal striatum; OFC, orbitofrontal cortex; VS, ventral striatum.
(TIF)

**S1 Text. Supporting information Tables A through F.** Numerical and statistical results related to Figs 5, 6 and 7. Data: https://doi.org/10.6084/m9.figshare.13139450.v1
(DOCX)

## Author Contributions

**Conceptualization:** R. Becket Ebitz, Benjamin Y. Hayden.

**Data curation:** R. Becket Ebitz, Jiaxin Cindy Tu.

**Formal analysis:** R. Becket Ebitz, Jiaxin Cindy Tu.

**Funding acquisition:** R. Becket Ebitz, Benjamin Y. Hayden.

**Investigation:** R. Becket Ebitz, Jiaxin Cindy Tu.

**Methodology:** R. Becket Ebitz, Jiaxin Cindy Tu.

**Project administration:** R. Becket Ebitz, Benjamin Y. Hayden.

**Resources:** R. Becket Ebitz, Benjamin Y. Hayden.

**Software:** R. Becket Ebitz, Jiaxin Cindy Tu.

**Supervision:** R. Becket Ebitz, Benjamin Y. Hayden.

**Validation:** R. Becket Ebitz, Jiaxin Cindy Tu.

**Visualization:** R. Becket Ebitz, Jiaxin Cindy Tu.

**Writing – original draft:** R. Becket Ebitz, Jiaxin Cindy Tu, Benjamin Y. Hayden.

**Writing – review & editing:** R. Becket Ebitz, Jiaxin Cindy Tu, Benjamin Y. Hayden.

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
