## [Editor Report · Decision Letter 0]

7 Feb 2020

Dear Dr Ebitz, 

Thank you for submitting your manuscript entitled "Rule adherence warps decision-making" for consideration as a Research Article by PLOS Biology.

Your manuscript has now been evaluated by the PLOS Biology editorial staff, as well as by an Academic Editor with relevant expertise, and I am writing to let you know that we would like to send your submission out for external peer review.

Please re-submit your manuscript within two working days, i.e. by Feb 11 2020 11:59PM.

Kind regards,

Gabriel Gasque, Ph.D.,

Senior Editor

PLOS Biology

---

## [Decision Letter · Decision Letter 1]

26 Mar 2020

Dear Dr Ebitz,

Thank you very much for submitting your manuscript "Rule adherence warps decision-making" for consideration as a Research Article at PLOS Biology. Your manuscript has been evaluated by the PLOS Biology editors, by an Academic Editor with relevant expertise, and by three independent reviewers. A fourth reviewer had also agreed to review your manuscript. However, this referee is now significantly overdue. Thus, not to delay communicating the decision any longer, we have decided to move forward with the comments we have in hand. If the fourth reviewer belatedly send us the review, we will forward it to you.

In light of the reviews (below), we will not be able to accept the current version of the manuscript, but we would welcome re-submission of a much-revised version that takes into account the reviewers' comments. We cannot make any decision about publication until we have seen the revised manuscript and your response to the reviewers' comments. Your revised manuscript is also likely to be sent for further evaluation by the reviewers.

We expect to receive your revised manuscript within 2 months. 

**IMPORTANT - SUBMITTING YOUR REVISION**

Your revisions should address the specific points made by each reviewer. As you will see, the reviewers raise various concerns regarding the manner in which you have interpreted the data, in particular questioning how well the data support the idea that rule implementation involves an attentional gate that warps how choices are represented in neural circuits. While we feel that it is appropriate to give you a chance to address the concerns raised by the reviewers, any final call on this study would depend on your ability to convince the reviewers that your interpretation of the data is solid.

Please submit the following files along with your revised manuscript:

*Re-submission Checklist*

*Published Peer Review*

*PLOS Data Policy*

*Blot and Gel Data Policy*

Sincerely,

Gabriel Gasque, Ph.D., 

Senior Editor

PLOS Biology

REVIEWS:

Reviewer #1: 

To start with I would like to apologise to authors and editors that this review was submitted a couple of days over deadline, a delay caused by an unforeseen family event. 

Ebitz, Tu & Hayden present data to test the hypothesis that the implementation of arbitrary stimulus response rules occurs through a mechanisms of attentional gating through which representations within decision making circuits are modified. They use neural recordings from OFC and striatum of macaques to argue that the presence of a rule in a decision-making paradigm modified the way in which potentially to-be-chosen items were represented in these regions. They use representational similarity analysis and choice predictive subspace analysis to show that rule relevant coding dimensions dominated over rule irrelevant dimensions in pseudopopulations of recorded cells, thereby supporting a hypothesis that there is a warping of representations in decision-making circuits specifically when a rule is being applied. 

There is much to like in this manuscript, it is largely well written and clear, and the analytic approach is of interest. I think there are some conceptual issues to overcome, however. 

Major Point

An interesting and I think useful feature of the author's approach is the use of the HMM to infer on the basis of the monkeys' responses which SR rule the monkey was currently using. 

This permits a reasonable inference of which of the designed task rules is currently being used and is a clear improvement on using the rule in force in the task as the analysing criterion. 

But of course the performance rules that the HMM can infer are effectively hard coded into the model, and the contrast is a default position of "rule-free" selection. The interpretation of the neural data relies heavily on this distinction. The evidence suggests that what is going on is more complex than this, and also there are alternative interpretations for the results presented that have not been considered. My major points are therefore about the cognitive context of the study and the resulting interpretation, rather than the neural data, but I think they are fairly fundamental.

1. The "rule-free" performance can be explained by a Win-Stay Lose-Shift rule, and so is not "rule free". 

Something is driving performance on "rule-free" trials, as despite being "rule-free" they are nevertheless significantly correct (1e), and in many cases this is because monkeys received trial feedback that the rule in force had changed, and then switched to a novel response. Indeed fig 1f clearly shows that this is the pattern for all "rule-free" choices. This is described in detail lines 151-162. This behaviour would appear, however, to be anything but "rule-free" - rather it adheres to the well-studied concept of "win-stay, lose-shift" rules, specifically in this case and exploratory Lose-Shift response. 

WSLS is well established in the monkey behaviour literature since Harlow argued that it drives classical learning set (see also Murray & Gaffan 2006 J Exp Psychol Anim Behav Process). It is also strongly established in general cognition and AI contexts (e.g. Nowak, M., & Sigmund, K. (1993). Nature, 364(6432) ). 

Although WSLS is not an SR rule like those coded into the HMM, it is a performance rule, and it would appear to be driving behaviour on the trials in question. Whilst there are alternative explanations in terms of performance rules, my general point is that it makes no sense to refer to these trials as "rule-free". 

The monkeys in this study have a vast cognitive training history (to quote from lines 637-9, "two types of foraging tasks, intertemporal choice tasks, two types of gambling tasks, attentional tasks, and a basic form of a reward-based decision task"). Many (admittedly not all) of these tasks benefit from a WSLS strategy. Indeed in many contexts WSLS is a "simpler" fall-back rule for monkeys, either when they are unable to capture a more complicated rule-set, or when reward rates mean that acquiring more complex strategies is not worth the cost. 

So maybe the authors need to consider more what is driving behaviour in these "rule-free" trials, and/or incorporate something like WSLS into their behavioural / model analysis. 

2. An explore-exploit trade-off also explains the neural data 

There is another way of looking at these data in terms of an explore-exploit trade-off - what the HMM seems to be distinguishing is trials where the monkey is exploiting (an SR rule), vs trials which fig 1f clearly suggests are exploratory, but termed "rule-free". 

Explore-exploit switches are known to lead to a state modulation of frontal activity in monkeys like that demonstrated in figure 2, indeed the first and senior authors have previously shown this in other cortical areas (e.g. Ebitz et al 2018 Neuron; Pearson et al 2009 Curr Biol, see also Quilodran et al 2008 Neuron). There are fewer data from the regions recorded here, but cf Costa & Averbeck 2020 J Neurosci. 

These are established findings in the literature, and I think they provide a reasonable explanation for the initial findings, for example fig 2. So how that viewpoint interacts with the very rule-centred interpretation of the current data is critical. This (and indeed the section above) would also call into question the central claim that rule-based decisions make more efficient use of neural resources. 

3. Terminology

There must be a difference between a state of responding on the basis of an inferred rule as decided by an HMM, a state of responding on the basis of an instructed/cued rule, and a state of responding based on a default acquired performance rule (c.f. learning set). The study as currently couched only draws on the first of these, yet throughout general conclusions are drawn about rule use. Some terminological clarification about the use here of the term "rule" is warranted. 

On a more general cognitive level, I would suggest that characterising behaviour as simply "rule-free", especially when performance is adaptive or high, doesn't make an awful lot of sense. The animals are making directed responses, this behaviour is thus driven by something. I accept that it's not being driven by a specific SR rule imposed in the task, but monkeys (and humans) don't simply respond using the rules we wanted to impose. 

So in summary I think the authors must consider these points in the context of what is nevertheless interesting data and a useful and well established task. I would actually argue that there is something much more interesting going on than the very simple story the authors present, related to context-driven switches between types of rule. But of course my role here is not tell the authors about the study I would have done with their nice data. 

Other points to consider

A number of comments are made throughout about reducing the energetic costs through the use of rules, though a link to the data is not really made explicit. I would imagine that the authors are referring to the reduced overall firing rate, but at the same time the encoding of rule-relevant dimensions seems to increase more than the compression of irrelevant dimensions. Is the argument here that the first effect has energetic consequences and not the second? If so, why? 

The suggestion (lines 70-72) that refs 12 and 13 fail to find evidence for an attentional gating hypothesis is, I think, not the completely correct characterisation of these studies. But they do propose alternative mechanisms, and it would be worth the authors discussing how these mechanisms can be integrated with the mechanisms they reveal. 

Fig 3 and associated text lines 215-222. The concept of "automatic or deliberative decision-making regions" as driving a very specific hypothesis to be tested is unclear to me. Whilst the idea is described in the introduction (87-96), what is not clear (either from the text or from the cited references) is how one would define an entire region as involved in deliberative decision making. As posed, this distinction feels a little phrenological, and I'm not sure this is how the authors mean it. If it is to be used, I feel a strong, formal, and evidence-based definition of what makes a whole area "deliberative" is necessary, along with a justification for the classification of whole brain areas as one or the other. 

Perhaps related, I find the "handoff" hypothesis under specified with respect to the supposedly contrasting warping hypothesis. The assertions that the evidence support warping are well justified, the parallel dismissal of handoff is much less clear - what would be the true neural correlates of this hypothesis, and do the authors really have the neural data (notably enough cortical regions) to test it? More generally, and related to the comments at the start, the idea of binary opposing hypotheses feels a little forced, when there is so much going on. 

Fig 4. Is pretty confusing. It doesn't help that there is no E) on the figure but there is in the legend (it appears that the current C) should be both C) and D)), and that the text (e.g. line 390) refers to 4C in the context of "identity-category index" when these results appear to be presented in 4D. Further, the legend could be much clearer, and all this made the figure really hard to interpret. Indications of the significant effects claimed in the text would also be helpful. 

Line 844-846 "All reported pseudopopulation results come from a single pseudopopulation, but were confirmed by bootstrap tests across 1000 randomly re-seeded pseudopopulations." Can we see the outcome of these bootstraps to have an idea of across pseudopopulation variation? Unless I have misunderstood something, this is not currently presented. 

Figure comments. 

1e and 1f would be clearer as boxplots or similar. Notably in 1e the comparison of red and grey is visually misleading. 

S1b would also be better presented as boxplots and it would be helpful to have axes that permit direct comparison between the 3 plots…

Typos

L77 "requires"

L97 we?

L756 the?

Reviewer #2: In this paper, Ebitz and colleagues present a study in monkeys learning stimulus-response mappings with reversals. A nice feature of the task is that three alternatives are provided for each choice dimensions, whereas many previous studies have focused on binary choices. Neuronal recordings are analyzed though little information is provided for motivating a focus on these particular regions. The paper overall is clearly written. I present below a few general comments, and then I develop more specific comments.

General comments

The authors present evidence that they interpret as consistent with a "warping hypothesis". However too often, it is not explained or demonstrated why the findings invalidate or are inconsistent with the alternative "handoff hypothesis", which seems to me not enough to support the claims. Otherwise both hypotheses could be consistent with the findings. One exception is Fig 3C, but it is quite an aggregate metric, could the authors provide more of such contrasts? Or rewrite the conclusions appropriately?

In particular the main finding that rule-relevant dimensions under rule-based, if I have understood correctly, does not seem to me contradictory with a handoff hypothesis. It remains to be demonstrated whether the results would also be compatible with an alternative explanation, for instance attentional gating that the authors touch upon briefly in their discussion.

Because only three regions were recorded, the title and conclusions seems to me too affirmative and general. I appreciate the value of such recordings, and the practical constraints, but it remains possible that other areas would be compatible with an alternative handoff explanation, as has been proposed before (this discussion point could be further developed). Because the three areas show similar patterns on most results presented, it remains possible that other areas would show patterns consistent with a handoff hypothesis instead. How were these recording sites selected in the first place?

Regarding the efficiency argument, while the authors present evidence that making choices under a rule require less resources (the experimenters observe less spikes) than on rule-free trials, what about the resources dedicated to the acquisition of the rule i.e. learning stimulus-response mappings in the first place? Could the authors explain why they do not take into account the neural resources to encode the rule per se that in turn will save resources when responding? Could the rule be learnt in the initial blocks or even training, and then subjects retrieve them when appropriate?

As I caveat I should say that because a lot of references are made to previous papers in terms of experimental paradigm, computational model, behavioral analyses, I could not review these in detail; having not worked with pseudopopulations myself, I am not able to appreciate whether this is a relevant approach here, though it does seem appropriate and it is well introduced and justified indeed by the authors.

Specific comments

Introduction

The introduction could be clearer regarding what is the question addressed: is it warping vs. handoff? Is it about testing the idea that warping is an efficient use of resources? It is unclear how this notion of efficiency is defined, and as compared to what.

The last part of the introduction does not explain why it is critical to compare following a rule when it is the correct rule vs. when it is not the correct rule. Could this be conflated by amount of reward received (more reward received when you follow the correct rule)?

Results

It remains possible that other computational models would also match the choice features very well. The model predicts rule-free trial after reversals, as expected when subjects are figuring out the new rule. It means that subjects' learning curves are very fast, with most their choices matching the true underlying state. Could the authors provide justification for their HMM approach, and explain why there is no model comparison?

Could the authors further quantify the distribution of rule-free choices instead of giving only some percentages: they are a third of all trials. Out of that third, how many are after a reversal (5 first trials of a block) or in a stable phase (rest of the block, as defined by the authors)? Out of these, how many correspond to correct vs. incorrect trials? Statements like « very often » are not enough, it is important to provide quantitative findings for readers to get a better sense of the behavior.

The statement that rule-free decisions by the model are actually « strategic, deliberative decisions » and not errors has to be demonstrated. Could the authors better explain why rule-free decisions being correct 49% of the time is inconsistent with a mixture of random choices and errors? It is also not surprising that animals are above chance after a reversal of rule, because rewards indicate that the current rule is no longer valid via reward feedback, so they switch to a new rule. But the chance level then is not really 33% correct, because subjects are now choosing between the rest of the alternative rules i.e. hypothesis testing (all minus the rule they have just switched away from)?

Instead of a deliberative strategy, could it be a simple pavlovian avoidance of choice features that were relevant before towards new choice features? The fact that monkeys made that choice often is not enough evidence that they engage in a strategic, deliberative thinking. Relatedly in Fig 1F, is the strategy different from a simple win stay lose shift?

Beginning of page 5: it is unclear if the comparison bears on all trials or only on the first trial after a change.

I disagree with or have misunderstood the authors' reasoning that after an error, it is a strategic choice to change both color and shape features at the same time: because if the choice is then rewarding, it will be difficult to infer whether it is due to the shape or to the color (credit assignment problem). Instead, if the subject changes one feature at a time, it can test whether that feature is relevant (as the authors nicely show in the mutual information inset). Isn't it incorrect to label these choices as « optimal », because it depends on what is actually optimal: deciding as fast as possible? Number of rewards obtained?

p. 5: « Rule-free trials in general were strongly biased towards… » again no quantitative information or statistical test is provided to support this statement.

It would be useful to look at the information on Fig 1G not on average but for trials following a block change. It is not demonstrated how such choices maximize reward too, not only information gain. Could the authors show whether the distinction rule-based/rule-free does not simply reduce to a « after reversal »/« plateau phase », just in terms of correctness, following the rule of the current block vs. not?

Could the authors present the results in Fig 3D separately for automatic and deliberative choices? Because most choices in the block will be automatic ones, they are the not discriminative choices for arbitrating between the two hypotheses.

Could another explanation for the discrepancy with fMRI findings is that the authors have recorded from a limited number of areas (understandably) as compared to whole brain measurements?

In Fig 4B does the conclusion boils down to the fact that after a reversal, more uncertainty so RSA more scattered, then more compact for rule-based when contingencies have been learnt? Could the authors discuss the lack of OFC modulation in rule-free choices: is OFC selective for rule-based as a cognitive map of task space (Wilson et al. 2014)?

In terms of writing, it would be useful if the authors further unpack their reasoning from the results to the implications or conclusions and claims that they derive. There was often a bit of a gap or a large step between the finding and a general conclusion or claim that followed.

Discussion

Could the authors engage more with the alternative explanations that they propose for their findings, in particular an attentional gain, and what would be the role of the other areas mentioned (dACC, DLPFC)?

How do these results relate to the identified dissociation between lateral PFC, medial PFC, OFC and striatum identified in humans and non-humans for rule-based and rule-free trials (e.g. Collins & Franck, 2014; Walton et al., 2010)?

Could the authors address rule acquisition (cf. general comment), at least comment on or speculate how this transition may occur? If space permits the discussion should also comment on the behavior and the model, especially in relation to extensive previous work on set acquisition and exploration.

Minor comments

Even if the HMM has been described elsewhere, it would be useful to have a list or a graph depiction of the Markov process and relevant variables

Could the authors indicate, out of all neurons recorded regardless of task properties, what is the proportion that present activity related to any task property (for each area)?

Fig 2a, could the authors add coordinates info regarding recording sites

Figure legends: could the authors keep « bars » for bars and indicate « error bars » or « shaded areas » where appropriate

« Was due to » p. 13 no correlation/causation is proved

A « transmission matrix » should be a transition matrix (HMM section)

Fig 4B / 5A color scale as legend should be symmetric otherwise the color indicating 0 is misleading

Mutual information: clarify methods section: « Two firing rate bins per neuron were used because maximized the number of observations per condition » 

"While this idea makes intuitive sense" is not a very scientific statement

Fig 3B, please clarify in the legend what are the grey shades representing

Methods section, category tuning: shouldn't the noise be epsilon(i,k) for trial k? Please clarify.

Reviewer #3: Main comments:

This is a nice manuscript with some interesting and unique observations. In particular, the contrast between rule-based and rule-free modes is interesting and noteworthy. The differences in between the striatum and OFC are also quite noteworthy. In general, a very nice set of results that will be of interest to anyone interested in decision-making.

I wonder, however, how much some of the interpretations of the results are a little oversold. The authors conclude that rule-adherence "warps" neural activity. This is certainly the case but is it a new observation of a re-packaging of old observations? It seems to me that the warping can be explained by already observed phenomenon, specifically feature-based attention and dimensionality reduction (categorization). The authors allude to prior work on these phenomena but I, for one, could not get my head around how "warping" is different. I am open to being convinced otherwise but the authors need to address this explicitly.

Mind you, this does not take away from a number of unique and important new observations. The observation that the rule-free state is not just random guessing but is instead used maximize information gathering is nice. The fact that this rule-free state vs the rule-based state engages the OFC and striatum differently is an important observation. But the idea that there is attentional selection during rule-based behavior is not all that new, at least as disseminated here. I guess the bottom-line is that the authors need to draw a sharper contrast and make more explicit how these observations differ from prior work showing feature-based attention and categorization (dimensionality reduction) shown previously. 

Minor comments:

1. There are multiple t-tests used for comparisons. It is more elegant and proper to use one test like an ANOVA (or other non-parametric stats, if warranted) and then post-hoc contrasts to identify where the differences lie. That reduces Type 1 errors. As far as I can tell, the t-tests were not corrected for multiple comparisons, which exacerbates the problem.

2. The authors state that reducing neural activity tends to reduce information. This is not really true. First of all, neurons with tuning curves tend to convey more information when they are not at peak excitability. And there are examples in the literature of neurons firing more sparsely, yet being more finely tuned after learning.

---

## [Decision Letter · Decision Letter 2]

1 Oct 2020

Dear Dr Ebitz,

Thank you very much for submitting a revised version of your manuscript "Rules warp feature encoding in decision circuits" for consideration as a Research Article at PLOS Biology. This revised version of your manuscript has been evaluated by the PLOS Biology editors and by the original the Academic Editor and reviewers. You will note that reviewer 1, Charlie Wilson, has identified himself. Please accept my apologies for the delay in sending the decision below to you.

In light of the reviews, we are pleased to offer you the opportunity to address the remaining points from the reviewers in a revised version that we anticipate should not take you very long. The Academic Editor has also provided detailed feedback, which I have included below my signature together with the reviewer comments. 

When you re-submit, we will assess your revised manuscript and your response to the reviewers' comments and we may consult the reviewers again.

We expect to receive your revised manuscript within 1 month.

**IMPORTANT - SUBMITTING YOUR REVISION**

Your revisions should address the specific points made by the Academic Editor and by each reviewer. Please submit the following files along with your revised manuscript:

*Resubmission Checklist*

*Published Peer Review*

*PLOS Data Policy*

*Blot and Gel Data Policy*

Sincerely,

Gabriel Gasque, Ph.D.,

Senior Editor,

ggasque@plos.org,

PLOS Biology

REVIEWS:

Academic Editor: 

1. I agree with reviewer 1. The term "rule-free" is somewhere between confusing and nonsensical. I also find their rebuttal disingenuous: that WSLS is not an adequate description of the "rule-free" trials because the monkey strategy is more complex. The very existence of what they call "rule-free" trials shows that the monkeys' strategy is more complex than following the 6 ground truth rules, but it doesn't invalidate them!

They need to drop the "rule-free" terminology in favour of a more neutral term. What is special about those trials is that they can't be explained by the model the authors have chosen to use. That's all. In other words, they are most likely a grab-bag of things that the monkey is doing that includes some idiosyncratic randomness, structured exploration, and heuristics like WSLS. This needs to be acknowledged. I suggest using the term "idiosyncratic" or "residual" or some such epithet which does not make strong claims about what is going on there. The authors are painting themselves into a corner with the term "rule free" because it obliged them to argue that, despite the fact that the monkeys are adopting a behavioural strategy that is quite effective, it's somehow not reaonably described as a "rule", which seems extremely odd.

2. Reviewer 2 originally pointed out that the the authors don't make a strong link between data and interpretation. When pushed to justify this, they simply removed one of the candidate explanations. I think a better way to deal with this is to honestly acknowledge any potential mismatch between data and theory. The reviewers are going to back this paper if the authors are straightforward, honest, and up-front about what their data do and do not say. So I would strongly suggest they orient the reader to clear, falsifiable predictions that either account would make for either hypothesis, and simply explain whether the data do (or do not) meet these. This would give the reader the opportunity to weigh up the evidence without excessive dressing up by the authors. 

Two comments of my own which seem substantive and were not raised at review, but which would set my mind at rest:

3. The authors see overall lower spike rates in the rule-based condition (when the monkey receives more positive outcomes) in three regions known to code for expected value . Is this a possible explanation?

4. Can the authors confirm that the RSA results were done in crossvalidation and avoid temporal confounds (neural signals for "rule-free" and "rule-based" are more similar within than between that designation because rule-free trials happen closer in time to other rule-free trials etc).

Reviewer #1, Charlie R E Wilson: The authors have considered my comments and those of the other reviewers seriously, and I think they have made significant strides in clarifying and improving the manuscript. I also find their interpretation more measured than previously, the figures improved and the extra supplementary stuff helpful. Their alteration of the theoretical context of the paper (in particular in response to reviewer 2 who made the point much better than I did) is appropriate and improves the paper. I still think there is much to like in this work, and that the manuscript merits publication.

I have one outstanding point to make. Perhaps the authors will find this pernickety as it is a point of language precision, but in re-reading the much-improved manuscript this phrase bothers me even more: we do not agree on the use of the term "rule-free". 

Now, the manuscript is much clearer on what the authors mean by this phrase, and the authors have taken the time to write a thoughtful explanation of their arrival at this term. I also agree with their argument for the rejection of the term "exploratory" which I think we will probably agree is over-used, particularly in the context of monkey behaviour. It is much clearer that this collection of trials is being performed using adaptive rules or strategies, and that it is a heterogenous set of choices. The reasons to class these trials by a description of exclusion are clear. These are all vital improvements. But I'm afraid that the phrase "rule-free" is still misleading. We could presumably have a very interesting back-and-forth on this for the next few months, but to be brief I think there are two points that render the term *at best* ambiguous: 1. Many readers will indeed interpret "rule-free" to mean "random". 2. There are valid arguments that one should consider "performance rules" (like WSLS) to be rules, even if the authors wish to consider them strategies. The authors make clear several times that their full meaning here is "sensorimotor rule-free". We can all agree on this, so why not use this term throughout (perhaps "SM rule free"?) in order to be, as the authors note, crystal clear. 

I have comments about the authors' response to my comments on WSLS. On reflection I do very much take their point that it is really the SM rule-based trials that are the focus of the neurophysiological analyses of this study. I therefore agree that the brief comments on WSLS that they have added now suffice to cover the issue. I'm including these comments anyway for completeness. 

The authors say "Because choices were multidimensional and the monkeys used a more complex strategy, a simple WSLS strategy isn't sufficient to explain our data" - this is pretty much an empirical claim for which the authors could provide the relevant analysis - I think there are versions of WSLS that can account for this behaviour across dimensions, particularly focussing on the lose-shift as the authors point out. The nature of the WSLS response depends what the animals have learned to shift *on*, and I think this is what the authors are getting at by their reference to "higher order" WSLS. This is *particularly* relevant in the context of the long and complicated testing history of these animals, something I think we all need to think more about in this field. 

Reviewer #2: In their revision, Ebitz and colleagues have satisfactorily addressed most of my comments, and have made substantial efforts to clarify their paper. I have one remaining main comment, corresponding to my first original comment:

"The authors present evidence that they interpret as consistent with a "warping hypothesis". However too often, it is not explained or demonstrated why the findings are inconsistent with an alternative "handoff hypothesis", which seems to me not enough to support the claims. Otherwise both hypotheses could be consistent with the findings. In particular the main finding that rule-relevant dimensions under rule-based does not seem to me contradictory with a handoff hypothesis. It remains to be demonstrated whether the results would also be compatible with an alternative explanation."

It is too bad that the authors have made the decision to remove entirely the alternative "hand-off" hypothesis. This has left me wondering if these are actually post hoc explanations, not prior hypotheses to arbitrate between. Since the study hypotheses were not preregistered, the authors should show where the evidence is not only compatible with the warping hypothesis, but also incompatible with other (at least, the hand-off) hypotheses. In the current version, the authors have simply discarded alternative hypotheses, and only evidence supporting the authors' hypothesis is presented, which could be compatible with other hypotheses too. I think that even without fully going back to the previous version around these hypotheses, it is possible to include these pieces of evidence where appropriate in the Results section, and some discussion about it.

And two remaining small comments:

- The authors have now included details about the anatomical landmarks for selecting their recording sites, but we are still told little about their motivation for focusing on these three regions, apart that they are broadly involved in decision-making (… like many other regions):

"We know from many studies that rule identity is encoded in firing rate changes of neurons in specific brain regions, classically in the dorsolateral prefrontal cortex, but also in regions implicated in decision-making, like the orbitofrontal cortex (OFC) and striatum." (introduction)

The paper would benefit from further justification here, especially as other (likely critical) areas are mentioned in their discussion.

- The authors often resort to very general terms like "intelligent", "clever" or "strategic", that can mean many things, and they do not specifically define what they mean here. For instance a WSLS can be strategic in certain contexts, as acknowledged by the authors. I recommend sticking to specific, precise language eg. « functions of the color, shape, and outcome of previous choices versus rule-based choices. ».

Reviewer #3: The authors have addressed my concerns.

---

## [Editor Report · Decision Letter 3]

8 Oct 2020

Dear Dr Ebitz,

Thank you for submitting your revised Research Article entitled "Rules warp feature encoding in decision circuits" for publication in PLOS Biology. I have now discussed your latest version with other staff editors and with the Academic Editor as well. We're delighted to let you know that we're now editorially satisfied with your manuscript. 

However, we would like to suggest a slight change to your title: "Rules warp feature encoding in decision-making circuits"

In addition, before we can formally accept your paper and consider it "in press", we also need to ensure that your article conforms to our guidelines. A member of our team will be in touch shortly with a set of requests. As we can't proceed until these requirements are met, your swift response will help prevent delays to publication. Please also make sure to address the data and other policy-related requests noted at the end of this email.

- a cover letter that should detail your responses to any editorial requests, if applicable

*Copyediting*

*Published Peer Review History*

*Early Version*

Sincerely,

Gabriel Gasque, Ph.D.,

Senior Editor,

ggasque@plos.org,

PLOS Biology

DATA POLICY:

Note that we do not require all raw data. Rather, we ask for all individual quantitative observations that underlie the data summarized in the figures and results of your paper. For an example see here: http://www.plosbiology.org/article/info%3Adoi%2F10.1371%2Fjournal.pbio.1001908#s5

These data can be made available in one of the following forms:

Regardless of the method selected, please ensure that you provide the individual numerical values that underlie the summary data displayed in the following figure panels: Figures 1D-G, 2BC, 3B-D, 4A-E, 5B-D, 6B-E, 7B-E, 8AB, S1, S2, S3, S4, and S5.

Please also ensure that each figure legend in your manuscript includes information on where the underlying data can be found and that your supplemental data file/s has/have a legend.

---

## [Editor Report · Decision Letter 4]

2 Nov 2020

Dear Dr Ebitz,

On behalf of my colleagues and the Academic Editor, Christopher Summerfield, I am pleased to inform you that we will be delighted to publish your Research Article in PLOS Biology. 

PRODUCTION PROCESS

Before publication you will see the copyedited word document (within 5 business days) and a PDF proof shortly after that. The copyeditor will be in touch shortly before sending you the copyedited Word document. We will make some revisions at copyediting stage to conform to our general style, and for clarification. When you receive this version you should check and revise it very carefully, including figures, tables, references, and supporting information, because corrections at the next stage (proofs) will be strictly limited to (1) errors in author names or affiliations, (2) errors of scientific fact that would cause misunderstandings to readers, and (3) printer's (introduced) errors. Please return the copyedited file within 2 business days in order to ensure timely delivery of the PDF proof. 

If you are likely to be away when either this document or the proof is sent, please ensure we have contact information of a second person, as we will need you to respond quickly at each point. Given the disruptions resulting from the ongoing COVID-19 pandemic, there may be delays in the production process. We apologise in advance for any inconvenience caused and will do our best to minimize impact as far as possible.

EARLY VERSION

PRESS 

Kind regards,

Alice Musson

Publishing Editor, 

PLOS Biology

on behalf of

Gabriel Gasque,

Senior Editor

PLOS Biology